# SOREL: A STOCHASTIC ALGORITHM FOR SPECTRAL RISKS MINIMIZATION

**Yuze Ge & Rujun Jiang** [*]
School of Data Science, Fudan University
`yzge23@m.fudan.edu.cn`
`rjjiang@fudan.edu.cn`

## ABSTRACT

The spectral risk has wide applications in machine learning, especially in real-world decision-making, where people are concerned with more than just average model performance. By assigning different weights to the losses of different sample points, rather than the same weights as in the empirical risk, it allows the model's performance to lie between the average performance and the worst-case performance. In this paper, we propose SOREL, the first stochastic gradient-based algorithm with convergence guarantees for spectral risks minimization. Previous approaches often rely on smoothing the spectral risk by adding a strongly concave function, thereby lacking convergence guarantees for the original spectral risk. We theoretically prove that our algorithm achieves a near-optimal rate of $\widetilde{O}(1/\sqrt{\epsilon})$ to obtain an $\epsilon$-optimal solution in terms $\epsilon$. Experiments on real datasets show that our algorithm outperforms existing ones in most cases, both in terms of runtime and sample complexity.

## 1 INTRODUCTION

In modern machine learning, model training heavily relies on minimizing the empirical risk. This ensures that the model have high average performance. Given a training set of $n$ sample points $\mathcal{D} = \{\boldsymbol{x}_i\}_{i=1}^n \subset \mathcal{X}$, the goal of the empirical risk minimization is to solve

$$\min_{\boldsymbol{w} \in \mathbb{R}^d} R(\boldsymbol{w}) = \frac{1}{n} \sum_{i=1}^n \ell_i(\boldsymbol{w}).$$

Here, $\boldsymbol{w} \in \mathbb{R}^d$ is the parameter vector of the model, $\ell_i(\boldsymbol{w}) = \ell(\boldsymbol{w}, \boldsymbol{x}_i)$ is the loss of the $i$-th sample, and $\ell : \mathbb{R}^d \times \mathcal{X} \to \mathbb{R}$ is the loss function. However, as machine learning models are deployed in real-world scenarios, the evaluation metrics for these models become more diverse, including factors such as fairness or risk aversion.

In this paper, we consider a generalized aggregation loss function: the spectral risk, which is of the form

$$R_{\boldsymbol{\sigma}}(\boldsymbol{w}) = \sum_{i=1}^n \sigma_i \ell_{[i]}(\boldsymbol{w}).$$

Here $\ell_{[1]}(\cdot) \leq \cdots \leq \ell_{[n]}(\cdot)$ denotes the order statistics of the empirical loss distribution, and $0 \leq \sigma_1 \leq \cdots \leq \sigma_n, \sum_{i=1}^n \sigma_i = 1$. In form, the spectral risk penalizes the occurrence of extreme losses by assigning greater weights to extreme losses. When $\sigma_i = 1/n$, the spectral risk reduces to the empirical risk. When $\sigma_n = 1$ and $\sigma_i = 0$ for $i = 1, \ldots, n-1$, the spectral risk becomes the maximum loss. Therefore, the spectral risk measures the model's performance between the average case and the worst case. By assigning different values to $\sigma_i$, the spectral risk encompasses a wide range of aggregated loss functions that have broad applications in fields such as machine learning and finance. Common spectral risks include Conditional Value at Risk (CVaR) or the average of top-k loss (Artzner, 1997; Rockafellar & Uryasev, 2000), Exponential Spectral Risk Measure (ESRM) (Cotter & Dowd, 2006), and Extremal Spectral Risk (Extremile) (Daouia et al., 2019). Their specific forms are shown in Table 1 (Mehta et al., 2022).

---

[*]Rujun Jiang is the corresponding author.

Table 1: Different spectral risks and the corresponding weights.

| Spectral Risks | Parameter | $\sigma_i$ |
|---|---|---|
| $\alpha$-CVaR | $0 < \alpha < 1$ | $\begin{cases} \frac{1}{n\alpha}, & i > \lceil n(1-\alpha) \rceil \\ 1 - \frac{\lfloor n\alpha \rfloor}{n\alpha}, & \lfloor n(1-\alpha) \rfloor < i < \lceil n(1-\alpha) \rceil \\ 0, & \text{otherwise} \end{cases}$ |
| $\rho$-ESRM | $\rho > 0$ | $e^{-\rho} \left( e^{\rho \frac{i}{n}} - e^{\rho \frac{i-1}{n}} \right) / \left(1 - e^{-\rho}\right)$ |
| $r$-Extremile | $r \geq 1$ | $\left(\frac{i}{n}\right)^r - \left(\frac{i-1}{n}\right)^r$ |

Despite the broad applications of spectral risks, optimization methods for spectral risks are still limited. In particular, for large-scale optimization problems, there is currently a lack of stochastic algorithms with convergence guarantees for the spectral risk minimization. Indeed, the weight of each sample point depends on the entire training set, introducing complex dependencies and thus making the optimization process challenging. Existing algorithms either abandon the convergence guarantee to the minimum of the spectral risk problem due to the difficulty of obtaining unbiased subgradient estimates (Levy et al., 2020; Kawaguchi & Lu, 2020), or turn to optimize the smooth regularized spectral risk (Mehta et al., 2024b; 2022), which lacks convergence guarantees for the original spectral risk. Given the widespread application of the spectral risk in machine learning and the lack of stochastic algorithms for the spectral risk minimization, we are committed to developing stochastic algorithms with convergence guarantees for the spectral risk minimization.

**Our Contributions.** In this paper, we propose the **S**tochastic **O**ptimizer for Spectral **R**isks minimization with traj**e**ctory Stabi**l**ization (SOREL). i) We propose SOREL, the first stochastic algorithm with convergence guarantees for the spectral risk minimization problem. In particular, SOREL stabilizes the trajectory of the primal variable to the optimal point. ii) Theoretically, we prove that SOREL achieves a near-optimal rate of $\widetilde{O}(1/\sqrt{\epsilon})$ to obtain an $\epsilon$-optimal solution in terms of the squared distance to the optimal point $\epsilon$ for spectral risks with a strongly convex regularizer. This matches the known lower bound of $\Omega(1/\sqrt{\epsilon})$ in the deterministic setting (Ouyang & Xu, 2021). iii) Experimentally, SOREL outperforms existing baselines in most cases, both in terms of runtime and sample complexity.

## 2 RELATED WORK

**Statistical Properties of the Spectral Risk.** As a type of risk measures, the spectral risk assigns higher weights to the tail distribution and has been profoundly studied in the financial field (Artzner et al., 1999; Rockafellar & Uryasev, 2013; He et al., 2022). Recently, statistical properties of the spectral risk have been investigated by many works in the field of learning theory. Specifically, Mehta et al. (2022); A. & Bhat (2022) demonstrate that the discrete form of spectral risks converges to the spectral risk of the overall distribution, controlled by the Wasserstein distance. Holland & Haress (2022); Khim et al. (2020); Holland & Haress (2021) also consider the learning bounds of spectral risks.

**Applications.** The spectral risk is widely applied in various fields of finance and machine learning. In some real-world tasks, the worst-case loss is as important as the average-case loss, such as medical imaging (Xu et al., 2022) or robotics (Sharma et al., 2020). The spectral risk minimization can be viewed as a risk-averse learning strategy. In the domain of fair machine learning, different subgroups are classified by sensitive features (e.g., gender and race). Subgroups with higher losses may be treated unfairly in decision-making. By penalizing samples with higher losses, the model's performance across different subgroups can meet certain fairness criteria (Williamson & Menon, 2019), such as demographic parity (Dwork et al., 2012) and equalized odds (Hardt et al., 2016). In the field of distributionally robust optimization (DRO), the sample distribution may deviate from a uniform distribution, which can be modeled by reweighting the samples (Chen & Paschalidis,

2020). Mehta et al. (2024b) adopts the spectral risk measures as the uncertainty set of the shifted distribution, which is similar to the form of the spectral risk minimization.

In practical applications, people can choose different types of spectral risks based on actual needs. For example, CVaR is popular in the context of portfolio optimization (Rockafellar & Uryasev, 2000), as well as reinforcement learning (Zhang et al., 2024; Chow et al., 2017). Levy et al. (2020) uses the CVaR measure as the uncertainty set in DRO, resulting in the same loss function as the spectral risk. Other applications of spectral risks include reducing test errors and mitigating the impact of outliers (Maurer et al., 2021; Kawaguchi & Lu, 2020; Fan et al., 2017), to name a few.

**Existing Optimization Methods.** There have been many algorithms to optimize CVaR, a special case of spectral risks, including both deterministic (Rockafellar & Uryasev, 2000) and stochastic algorithms (Fan et al., 2017; Curi et al., 2020). For the spectral risk, deterministic methods such as subgradient methods have convergence guarantees, although they are considered algorithms with slow convergence rate. Xiao et al. (2023) propose an Alternating Direction Method of Multipliers (ADMM) type method for the minimization of the rank-based loss, inspired by Cui et al. (2024). Other deterministic methods reformulate this problem into a minimax problem (Thekumparampil et al., 2019; Hamedani & Aybat, 2021; Khalafi & Boob, 2023). However, these methods require calculating $O(n)$ function values and gradients at each iteration, posing significant limitations when solving large-scale problems.

Stochastic algorithms for solving the spectral risk minimization problems are still limited. Some algorithms forgo convergence to the true minimum of the spectral risk (Levy et al., 2020; Kawaguchi & Lu, 2020). Other methods modify the objective to minimize a smooth approximation of the spectral risk by adding a strongly concave term with a coefficient $\nu$ (Mehta et al., 2022; 2024b). The smaller $\nu$ is, the closer the approximation is to the original spectral risk. Mehta et al. (2024b) propose the Prospect algorithm and prove that it achieves linear convergence for any $\nu > 0$. Furthermore, if the loss of each sample is different at the optimal point, then the optimal value of the smooth approximation of the spectral risk is the same as the optimal value of the original spectral risk as long as $\nu$ is below a certain positive threshold. However, in practice, these conditions are difficult to verify. The convergence of these algorithms still lacks guarantees for original spectral risks minimization. Other stochastic algorithms, including Hamedani & Jalilzadeh (2023); Yan et al. (2019), designed for solving general minimax problems, have a slower convergence rate of $O(1/\epsilon)$ in terms of $\epsilon$ to obtain an $\epsilon$-optimal solution.

After our original submission of this work we have also recently become aware that Mehta et al. (2024a) proposed a primal-dual stochastic algorithm to solve the DRO problem. Moreover, their algorithm can also be improved to minimize the original spectral risk, even though their convergence analysis focuses on the smoothed spectral risk. We discuss this in Appendix F in detail.

## 3 ALGORITHM

We consider stochastic optimization of the spectral risk combined with a strongly convex regularizer:

$$\min_{\boldsymbol{w}} \underbrace{\sum_{i=1}^{n} \sigma_i \ell_{[i]}(\boldsymbol{w})}_{R_{\boldsymbol{\sigma}}(\boldsymbol{w})} + g(\boldsymbol{w}). \tag{1}$$

Firstly, we make the basic assumption about the individual loss function $\ell_i$ and the regularizer $g$.

**Assumption 1** *The individual loss function $\ell_i : \mathbb{R}^d \to \mathbb{R}$ is convex, G-Lipschitz continuous and L-smooth for all $i \in \{1, \ldots, n\}$. The regularizer $g : \mathbb{R}^d \to \mathbb{R} \cup \{\infty\}$ is proper, lower semicontinuous and $\mu$-strongly convex.*

Assumption 1 is a standard assumption in the literature on stochastic optimization (Nemirovski et al., 2009; Davis & Drusvyatskiy, 2019), especially in the field of the spectral risk minimization (Kawaguchi & Lu, 2020; Holland & Haress, 2022; Levy et al., 2020; Mehta et al., 2022; 2024b). The logistic loss satisfies this assumption. The least-square loss satisfies this assumption as long as the iterative sequence lies in a bounded sublevel set. The assumption of strong convexity of $g$ is very common, for example, the $l_2$ regularization is widely used in machine learning.

### 3.1 CHALLENGES OF STOCHASTIC OPTIMIZATION FOR SPECTRAL RISKS

In this section, we describe the challenges of the spectral risk minimization problem and the techniques to solve them.

**Biases of Stochastic Subgradient Estimators.** From convex analysis (Wang et al., 2023, Lemma 10), we know that the subgradient of $R_{\boldsymbol{\sigma}}$ is

$$\partial R_{\boldsymbol{\sigma}}(\boldsymbol{w}) = \mathrm{Conv}\left\{\bigcup_{\pi}\left\{\sum_{i=1}^{n}\sigma_i \nabla\ell_{\pi(i)}(\boldsymbol{w}) : \ell_{\pi(1)}(\boldsymbol{w}) \leq \cdots \leq \ell_{\pi(n)}(\boldsymbol{w})\right\}\right\},$$

where $\mathrm{Conv}$ denotes the convex hull of a set, and $\pi$ is a permutation that arranges $\ell_1, \ldots, \ell_n$ in ascending order. Note that $R_{\boldsymbol{\sigma}}(\boldsymbol{w})$ is non-smooth. Indeed, when there exist $i \neq j$ such that $\ell_i(\boldsymbol{w}) = \ell_j(\boldsymbol{w})$, $\partial R_{\boldsymbol{\sigma}}(\boldsymbol{w})$ contains multiple elements.

The subgradient of $R_{\boldsymbol{\sigma}}$ is related to the ordering of $\ell_1, ..., \ell_n$. We cannot obtain an unbiased subgradient estimator of $\partial R_{\boldsymbol{\sigma}}$ if we use only a mini-batch with $m$ ($m < n$) sample points. For example, when $m = 1$, we randomly sample $i$ uniformly from $\{1, \ldots, n\}$. The subgradient estimator $\nabla\ell_i(\boldsymbol{w})$ is unbiased only if $\sigma_i = 1/n$. For general $\boldsymbol{\sigma}$, unfortunately, to obtain an unbiased subgradient estimator of $\partial R_{\boldsymbol{\sigma}}$, we have to compute $n$ loss function values and then determine the ranking of $\ell_i$ among the $n$ losses (or the weight corresponding to the $i$-th sample point). However, computing $O(n)$ losses at each step is computationally heavy. To remedy this, we next design an algorithm that first uses a minimax reformulation of Problem (1) and then alternately updates the weights of each sample point and $\boldsymbol{w}$ using a primal-dual method.

Equivalently, we can rewrite $R_{\boldsymbol{\sigma}}(\boldsymbol{w})$ in the following form

$$R_{\boldsymbol{\sigma}}(\boldsymbol{w}) = \max_{\boldsymbol{\lambda}\in\Pi_{\boldsymbol{\sigma}}}\sum_{i=1}^{n}\lambda_i\ell_i(\boldsymbol{w}), \tag{2}$$

where $\Pi_{\boldsymbol{\sigma}} = \{\Pi\boldsymbol{\sigma} : \Pi\mathbf{1} = \mathbf{1}, \Pi^{\top}\mathbf{1} = \mathbf{1}, \Pi \in [0,1]^{n\times n}\}$ is the permutahedron associated with $\boldsymbol{\sigma}$, i.e., the convex hull of all permutations of $\boldsymbol{\sigma}$, and $\mathbf{1}$ is the all-one vector (Blondel et al., 2020). Then Problem (1) can be rewritten as

$$\min_{\boldsymbol{w}}\max_{\boldsymbol{\lambda}\in\Pi_{\boldsymbol{\sigma}}}L(\boldsymbol{w},\boldsymbol{\lambda}) = \sum_{i=1}^{n}\lambda_i\ell_i(\boldsymbol{w}) + g(\boldsymbol{w}). \tag{3}$$

Next, we use a primal-dual method to solve Problem (3). Specifically, we iteratively update $\boldsymbol{w}$ and $\boldsymbol{\lambda}$:

$$\boldsymbol{\lambda}_{k+1} = \arg\max_{\boldsymbol{\lambda}\in\Pi_{\boldsymbol{\sigma}}}\sum_{i=1}^{n}\lambda_i\ell_i(\boldsymbol{w}_k) - \frac{1}{2\eta_k}\|\boldsymbol{\lambda} - \boldsymbol{\lambda}_k\|^2, \tag{4}$$

$$\boldsymbol{w}_{k+1} = \arg\min_{\boldsymbol{w}}P_k(\boldsymbol{w}) := \sum_{i=1}^{n}\lambda_{i,k+1}\ell_i(\boldsymbol{w}) + g(\boldsymbol{w}) + \frac{1}{2\tau_k}\|\boldsymbol{w} - \boldsymbol{w}_k\|^2. \tag{5}$$

Steps (4) and (5) can be seen as alternatingly solving the min problem and the max problem in (3) with proximal terms.

**Stabilizing the Optimization Trajectory.** To update $\boldsymbol{\lambda}_{k+1}$, one may naturally think of solving Problem (2): $\boldsymbol{\lambda}_{k+1} = \arg\max_{\boldsymbol{\lambda}\in\Pi_{\boldsymbol{\sigma}}}\sum_{i=1}^{n}\lambda_i\ell_i(\boldsymbol{w}_k)$, similar to methods in Mehta et al. (2022; 2024b) with smoothing coefficient $\nu = 0$. However, since Problem (2) is merely convex, the solution $\boldsymbol{\lambda}$ lacks continuity with respect to $\boldsymbol{w}$, that is, a small change in $\boldsymbol{w}$ could lead to a large change in $\boldsymbol{\lambda}$. Indeed, it is often the case that there are multiple optimal solutions for (2) when there exists $i \neq j$ such that $\ell_i(\boldsymbol{w}) = \ell_j(\boldsymbol{w})$, and in this case, an arbitrary small perturbation of $\boldsymbol{w}$ will lead to a different value of $\lambda_i$. As shown in Figure 1, this can cause $\boldsymbol{w}$ to oscillate near points where some losses are the same and prevents the convergence of the algorithm. We also provide a toy example in Appendix C to further illustrate this difficulty. Therefore, the proximal term $\frac{1}{2\eta_k}\|\boldsymbol{\lambda} - \boldsymbol{\lambda}_k\|^2$ is added in (4) to prevent excessive changes in $\boldsymbol{\lambda}$ and stabilize the trajectory of the primal variable, where $\eta_k > 0$ controls the extent of its variation.

Without Trajectory Stabilization                    With Trajectory Stabilization

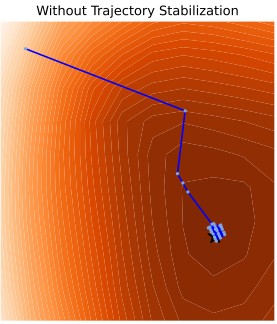 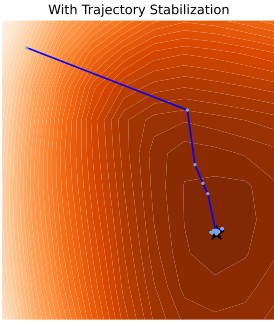

Figure 1: The level set plot of 2D least-square regression with primal-dual optimization trajectories described in Section 3.1. The max subproblem does not have a proximal term (**left**) or has a proximal term (**right**). The min subproblem does not have a proximal term. The black star represents the optimal point. The sample points are obtained by projecting the `yacht` dataset onto $\mathbb{R}^2$ using PCA.

**Stochastic Optimization for the Primal Variable.** We use a stochastic algorithm to approximately solve (5). Through the minimax reformulation in (5), we avoid directly calculating the stochastic subgradient of $R_{\boldsymbol{\sigma}}(\boldsymbol{w})$, which requires computing all loss function values to obtain the corresponding sample weight $\lambda_i$. Additionally, since $\boldsymbol{\lambda}_{k+1}$ is fixed, the finite sum part of the objective function in (5) is smooth, allowing us to use variance reduction (VR), a commonly used technique in stochastic optimization (Shalev-Shwartz & Zhang, 2013; Roux et al., 2012; Johnson & Zhang, 2013; Defazio et al., 2014), to accelerate our stochastic algorithm. In contrast, since $R_{\boldsymbol{\sigma}}(\boldsymbol{w})$ is non-smooth, as previously mentioned, VR cannot be used to directly solve Problem (1). For smooth convex functions in the form of the finite sum, many methods such as SVRG (Johnson & Zhang, 2013), SAGA (Defazio et al., 2014), and SARAH (Nguyen et al., 2017) can enable stochastic methods to achieve the convergence rate of deterministic methods. We apply the proximal stochastic gradient descent with a generalized VR method inspired by SVRG to approximately solve (5), which will be presented in Section 3.2 in detail. Thanks to its strong convexity, Problem (5) can be solved efficiently.

Similar to (4), we add a proximal term $\frac{1}{2\tau_k}\|\boldsymbol{w}-\boldsymbol{w}_k\|^2$ in (5) where $\tau_k > 0$ is the proximal parameter. The proximal parameter $\tau_k$ is crucial for the convergence proof of our algorithm. By carefully choosing $\tau_k = O(1/k)$, the updates of $\boldsymbol{w}$ become more stringent as the algorithm progresses, and SOREL can achieve a near optimal rate of $\widetilde{O}(1/\sqrt{\epsilon})$ in terms of $\epsilon$.

## 3.2 THE SOREL ALGORITHM

Our proposed algorithm SOREL is summarized in Algorithm 1. The specific values for the parameters $\theta_k, \eta_k, \tau_k$ and $m_k$ in Algorithm 1 will be given in Section 4. In Line 2 the algorithm initializes $\boldsymbol{\lambda}_0$ by solving Problem (2). In Lines 8-15, the algorithm computes the stochastic gradient and update $\boldsymbol{w}$ for a fixed $\boldsymbol{\lambda}$, as described in Section 3.1. Additionally, we compute the full gradient of $\boldsymbol{w}$ every $m_k$ updates to reduce the variance. In Lines 4-5, we update $\boldsymbol{\lambda}$. Note that we replace $\ell_i(\boldsymbol{w}_k)$ with $\ell_i(\boldsymbol{w}_k) + \theta_k \left(\ell_i(\boldsymbol{w}_k) - \ell_i(\boldsymbol{w}_{k-1})\right)$ to accelerate the algorithm. This can be seen as a momentum term, a widely used technique in smooth optimization (Tseng, 1998; Liu et al., 2020; Gitman et al., 2019; Sutskever et al., 2013), where $\theta_k > 0$ is the momentum parameter.

Define the proximal operator $\text{prox}_h(\bar{\boldsymbol{x}}) := \arg\min_{\boldsymbol{x}} h(\boldsymbol{x}) + \frac{1}{2}\|\boldsymbol{x} - \bar{\boldsymbol{x}}\|^2$ for a function $h$. In Line 15, we apply the proximal stochastic gradient descent step. We assume that $\text{prox}_{g+\frac{1}{2}\|\cdot\|^2}(\cdot)$ is easy to compute, which is the case for many commonly used regularizers $g$, such as the $l_1$ norm and the elastic net regularization (Zou & Hastie, 2005). If $g$ is differentiable, we can replace the proximal stochastic gradient with stochastic gradient: $\boldsymbol{w}_{k,t+1} = \boldsymbol{w}_{k,t} - \alpha\left(\boldsymbol{d}_{k,t} + \frac{1}{\tau_k}(\boldsymbol{w}_{k,t} - \boldsymbol{w}_k) + \nabla g(\boldsymbol{w}_{k,t})\right)$. This will not affect the convergence or convergence rate of the algorithm as long as $\nabla g$ is Lipschitz continuous and the step size $\alpha$ is small enough. In Line 5, we need to compute the projection onto $\Pi_{\boldsymbol{\sigma}}$. For an ordered vector, projecting onto the permutahedron takes $O(n)$ operations using the Pool Adjacent Violators Algorithm (PAVA)(Lim & Wright,

2016). In SOREL, we need to first sort $n$ elements of the projected vector and then compute the projection onto $\Pi_{\boldsymbol{\sigma}}$, which takes a total of $O(n \log n)$ operations.

In practice, we set $T_k$ and $m_k$ to $n$ in Lines 8 and 9, meaning the algorithm updates $\boldsymbol{\lambda}$ once it traverses the training set. We also set the reference point $\bar{\boldsymbol{w}}$ and the output $\boldsymbol{w}_{k+1}$ in Lines 10 and 17 to be the last vector of the previous epoch rather than the average vector, as with most practical algorithms (Johnson & Zhang, 2013; Zhu & Hazan, 2016; Cutkosky & Orabona, 2019; Babanezhad et al., 2015; Gower et al., 2020). In this way, SOREL only requires computing the full batch gradient once for each update of $\boldsymbol{\lambda}$, and becomes single-loop in Lines 8- 16. This makes the algorithm more concise and parameters easier to tune. Additionally, in the Appendix D, we provide the SOREL algorithm with mini-batching.

---

**Algorithm 1** SOREL

---

1: **Input:** initial $\boldsymbol{w}_0$, $\boldsymbol{w}_{-1} = \boldsymbol{w}_0$, $\boldsymbol{\sigma}$, and learning rate $\alpha$, $\{\theta_k\}_{k=0}^{K-1}$, $\{\eta_k\}_{k=0}^{K-1}$, $\{\tau_k\}_{k=0}^{K-1}$, $\{m_k\}_{k=0}^{K-1}$ and $\{T_k\}_{k=0}^{K-1}$.
2: $\boldsymbol{\lambda}_0 = \arg\min_{\boldsymbol{\lambda} \in \Pi_{\boldsymbol{\sigma}}} -\boldsymbol{\ell}(\boldsymbol{w}_0)^{\top} \boldsymbol{\lambda}$.
3: **for** $k = 0, \ldots, K-1$ **do**
4: $\quad$ $\boldsymbol{v}_k = (1+\theta_k)\boldsymbol{\ell}(\boldsymbol{w}_k) - \theta_k\boldsymbol{\ell}(\boldsymbol{w}_{k-1})$.
5: $\quad$ $\boldsymbol{\lambda}_{k+1} = \arg\min_{\boldsymbol{\lambda} \in \Pi_{\boldsymbol{\sigma}}} -\boldsymbol{v}_k^{\top}\boldsymbol{\lambda} + \frac{1}{2\eta_k}\|\boldsymbol{\lambda} - \boldsymbol{\lambda}^k\|^2$.
6: $\quad$ $\boldsymbol{w}_{k,0} = \boldsymbol{w}_k$, $\bar{\boldsymbol{w}} = \boldsymbol{w}_k$.
7: $\quad$ $\bar{\boldsymbol{g}} = \sum_{i=1}^n \lambda_{i,k+1}\nabla\ell_i(\bar{\boldsymbol{w}})$.
8: $\quad$ **for** $t = 1, \ldots, T_k$ **do**
9: $\quad\quad$ **if** $t \bmod m_k = 0$ **then**
10: $\quad\quad\quad$ $\bar{\boldsymbol{w}} = \frac{1}{m_k}\sum_{j=t-m_k+1}^t \boldsymbol{w}_{k,j}$.
11: $\quad\quad\quad$ $\bar{\boldsymbol{g}} = \sum_{i=1}^n \lambda_{i,k+1}\nabla\ell_i(\bar{\boldsymbol{w}})$.
12: $\quad\quad$ **end if**
13: $\quad\quad$ Sample $i_t$ uniformly from $\{1, \ldots, n\}$,
14: $\quad\quad$ $\boldsymbol{d}_{k,t} = n\lambda_{i_t,k+1}\nabla\ell_{i_t}(\boldsymbol{w}_{k,t}) - n\lambda_{i_t,k+1}\nabla\ell_{i_t}(\bar{\boldsymbol{w}}) + \bar{\boldsymbol{g}}$
15: $\quad\quad$ $\boldsymbol{w}_{k,t+1} = \text{Prox}_{\alpha\left(g + \frac{1}{2\tau_k}\|\cdot - \boldsymbol{w}_k\|^2\right)}\left\{\boldsymbol{w}_{k,t} - \alpha\boldsymbol{d}_{k,t}\right\}$.
16: $\quad$ **end for**
17: $\quad$ $\boldsymbol{w}_{k+1} = \frac{1}{m_k}\sum_{j=T_k-m_k+1}^{T_k} \boldsymbol{w}_{k,j}$.
18: **end for**
19: **Output:** $\boldsymbol{w}_K$.

---

## 4 THEORETICAL ANALYSIS

For convenience, we consider that $T_k$ (will be determined in Theorem 1) is large enough so that $\boldsymbol{w}_k$ is a $\delta_k$-optimal solution of $P_k(\boldsymbol{w})$, that is, $\mathbb{E}_k P_k(\boldsymbol{w}_{k+1}) - \min_{\boldsymbol{w}} P_k(\boldsymbol{w}) \leq \delta_k$. Here, $\mathbb{E}_k$ represents the conditional expectation with respect to the random sample points used to compute $\boldsymbol{w}_{k+1}$ given $\boldsymbol{w}_k, \ldots, \boldsymbol{w}_0$. Then, we can provide a one-step analysis of the outer loop of SOREL. We use $L(\boldsymbol{w}, \boldsymbol{\lambda}) = \boldsymbol{\lambda}^{\top}\boldsymbol{\ell}(\boldsymbol{w}) + g(\boldsymbol{w})$ in the analysis for simplicity. The cnvergence analysis for SOREL with mini-batching is presented in Appendix D.

**Lemma 1** *Suppose Assumption 1 holds. Let $\{\boldsymbol{w}_k\}$ and $\{\boldsymbol{\lambda}_k\}$ be the sequences generated by Algorithm 1. Then for any $\boldsymbol{w} \in \mathbb{R}^d$, $\boldsymbol{\lambda} \in \Pi_{\boldsymbol{\sigma}}$ and $D = G/\mu$, the following inequality holds,*

$$
\mathbb{E}_k \left\{ L(\boldsymbol{w}_{k+1}, \boldsymbol{\lambda}) - L(\boldsymbol{w}, \boldsymbol{\lambda}_{k+1}) \right\}
$$

$$
\leq \mathbb{E}_k \left\{ \langle \boldsymbol{\lambda} - \boldsymbol{\lambda}_{k+1}, \boldsymbol{\ell}(\boldsymbol{w}_{k+1}) \rangle + \frac{1}{2\eta_k}\left[\|\boldsymbol{\lambda} - \boldsymbol{\lambda}_k\|^2 - \|\boldsymbol{\lambda} - \boldsymbol{\lambda}_{k+1}\|^2 - \|\boldsymbol{\lambda}_{k+1} - \boldsymbol{\lambda}_k\|^2\right] \right.
$$

$$
+ \frac{1}{2\tau_k}\left[\|\boldsymbol{w} - \boldsymbol{w}_k\|^2 - \|\boldsymbol{w} - \boldsymbol{w}_{k+1}\|^2 - \|\boldsymbol{w}_{k+1} - \boldsymbol{w}_k\|^2\right] - \frac{\mu}{2}\|\boldsymbol{w} - \boldsymbol{w}_{k+1}\|^2 \tag{6}
$$

$$
\left. + \langle \boldsymbol{v}_k, \boldsymbol{\lambda}_{k+1} - \boldsymbol{\lambda} \rangle + \delta_k + \sqrt{\frac{(\tau_k^{-1} + \mu)\delta_k}{2}}(D^{-1}\|\boldsymbol{w} - \boldsymbol{w}_{k+1}\|^2 + D) \right\}.
$$

Next, we try to telescope the terms on the right hand side of (6) by multiplying each term by $\gamma_k$. By choosing appropriate parameters in Algorithm 1 to satisfy some conditions (will be discussed in Appendix A), we can ensure that the adjacent terms indexed by $k = 0, \dots, K - 1$ can be canceled out during summation. Then we can achieve the desired convergence result.

**Theorem 1** *Suppose Assumption 1 holds. Set $\gamma_k = k + 1$, $\eta_k = \frac{\mu(k+1)}{8nG^2}$, $\theta_k = \frac{k}{k+1}$, $\tau_k = \frac{4}{\mu(k+1)}$, $\delta_k = D^2 \min\left(\frac{\mu}{8(k+5)}, \mu(k+1)^{-6}\right)$, $D = G/\mu$, the step-size $\alpha = \frac{1}{12L}$, $m_k = \frac{384L}{(k+5)\mu} + 2$ and $T_k = O(m_k \log \frac{1}{\delta_k})$ in Algorithm 1. Let $\boldsymbol{w}^\star$ be the optimal solution of Problem (1). Then we have $\mathbb{E}\|\boldsymbol{w}_K - \boldsymbol{w}^\star\|^2 = O\left(\frac{nG^2}{\mu^2 K^2}\right)$.*

**Corollary 1** *Under the same conditions in Theorem 1, we obtain an output $\boldsymbol{w}_K$ of Algorithm 1 such that $\mathbb{E}\|\boldsymbol{w}_K - \boldsymbol{w}^\star\|^2 \le \epsilon$ in a total sample complexity of $O\left(\frac{n^{\frac{3}{2}}G}{\mu\sqrt{\epsilon}} \log \frac{1}{\epsilon} + \frac{L}{\mu}\left(\log \frac{1}{\epsilon}\right)^2\right)$.*

Our algorithm achieves a near-optimal convergence rate of $\widetilde{O}(1/\sqrt{\epsilon})$ in terms of $\epsilon$, which matches the lower bound of $\Omega(1/\sqrt{\epsilon})$ in the deterministic setting up to a logarithmic term (Ouyang & Xu, 2021). This is the first near-optimal stochastic method for solving the spectral risk minimization problem. Previously, Mehta et al. (2022; 2024b) add a strongly concave term with respect to $\boldsymbol{\lambda}$ in $L(\boldsymbol{w}, \boldsymbol{\lambda})$ and achieve a linear convergence rate for the perturbed problem. One may set the coefficient of the strongly concave term $\nu$ to $O(\epsilon)$, obtaining an $\epsilon$-optimal solution for the original spectral risk minimization problem. However, this approach has drawbacks: it leads to a worse sample complexity of $\widetilde{O}(1/\epsilon)$ (Palaniappan & Bach, 2016) or even $\widetilde{O}(1/\epsilon^3)$ (Mehta et al., 2024b); additionally, to achieve an $\epsilon$-optimal solution, the step size would need to be set to $O(\epsilon)$, resulting in very small steps that perform poorly in practice. In contrast, SOREL's step size is independent of $\epsilon$. We also discuss the dependence of complexity on $n$ in Appendix F.

## 5 EXPERIMENTS

In this section, we compare our proposed algorithm SOREL with existing baselines for solving the spectral risk minimization problem. In addition to the precision of the optimizers during training, we also explore fairness and distribution shift metrics on the test set. We focus more on the performance of an optimizer during the training process; therefore, we do not pursue state-of-the-art test metrics due to potential overfitting issues.

We train linear models with $l_2$ regularization in all experiments. We adopt a wide variety of spectral risks, including ESRM, Extremile, and CVaR. Baseline methods include SGD (Mehta et al., 2022) with a minibatch size of 64, LSVRG (Mehta et al., 2022), and Prospect (Mehta et al., 2024b). Note that although both LSVRG and Prospect add a strongly concave term with coefficient $\nu$ to smooth the original spectral risk, they have been observed to exhibit linear convergence for the original spectral risk minimization problem in practice without the strongly concave term (Mehta et al., 2022; 2024b). Consequently, we set $\nu = 0$ in our experiments. Detailed experimental settings are provided in Appendix B. The source code is available at `https://github.com/SXFXuz/SOREL`.

### 5.1 LEAST-SQUARES REGRESSION

Five tabular regression benchmarks are used for the least squares loss: `yacht` (Tsanas & Xifara, 2012), `energy` (Baressi Šegota et al., 2019), `concrete` (Yeh, 2006), `kin8nm` (Akujuobi & Zhang, 2017), `power` (Tüfekci, 2014). We compare the suboptimality versus passes (the number of samples divided by $n$) and runtime. The suboptimality is defined as

$$\text{Suboptimality}(\boldsymbol{w}_k) = \frac{R_{\boldsymbol{\sigma}}(\boldsymbol{w}_k) + g(\boldsymbol{w}_k) - R_{\boldsymbol{\sigma}}(\boldsymbol{w}^\star) - g(\boldsymbol{w}^\star)}{R_{\boldsymbol{\sigma}}(\boldsymbol{w}_0) + g(\boldsymbol{w}_0) - R_{\boldsymbol{\sigma}}(\boldsymbol{w}^\star) - g(\boldsymbol{w}^\star)},$$

where $\boldsymbol{w}^\star$ is calculated by L-BFGS (Nocedal & Wright, 1999).

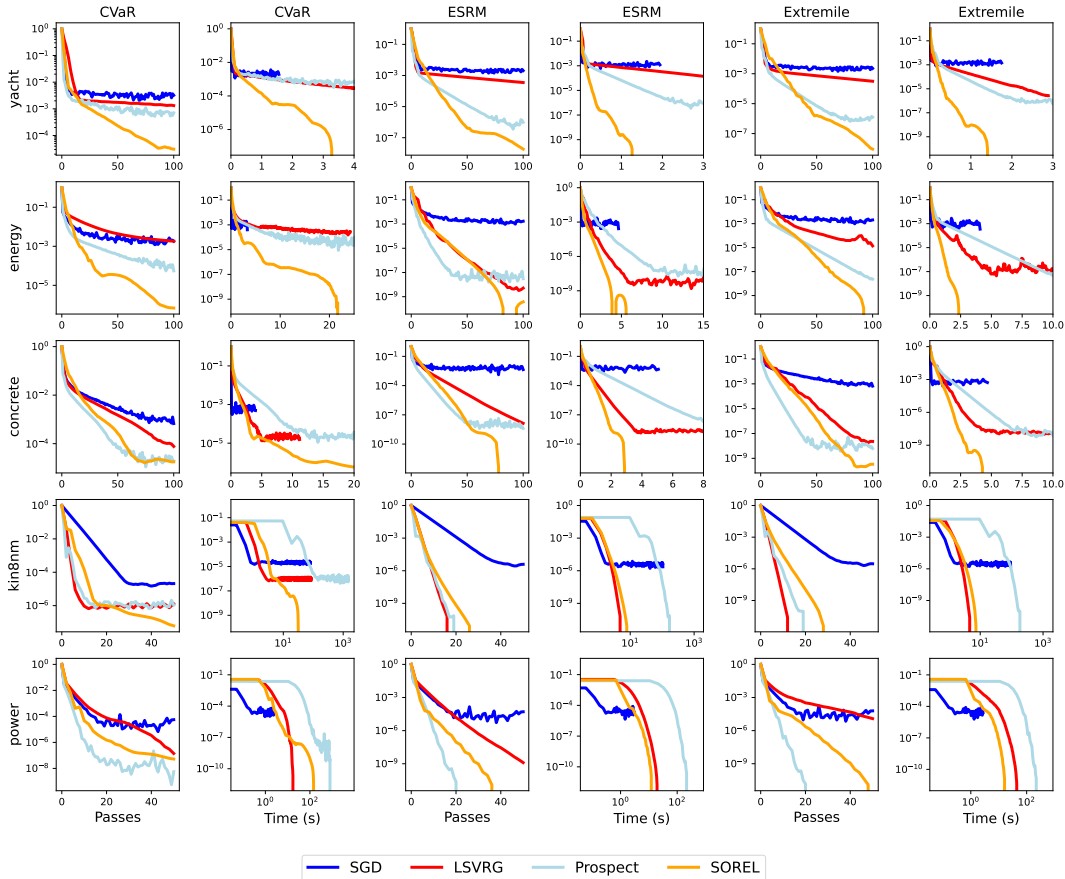

Figure 2: Suboptimality of spectral risks for different algorithms **without mini-batching**. The $x$-axis represents the effective number of samples used by the algorithm divided by $n$ (odd columns) or CPU time (even columns). Each row corresponds to the same dataset, and each column corresponds to the same type of the spectral risk.

**Results.** Figure 2 compares the training curves of our method with other baselines across various datasets and the spectral risk settings. In terms of sample complexity and runtime, SOREL outperforms other baselines in most cases; SOREL also achieve comparable results in the `kin8nm` dataset. In the `power` dataset, the sample complexity of Prospect is better than that of SOREL. However, the runtime of SOREL is significantly shorter than that of Prospect due to the fact that Prospect needs the calculation of projections onto the permutahedron with $O(n)$ operations each step. As expected, SGD fails to converge due to its inherent bias (Mehta et al., 2022). Although Mehta et al. (2024b) discusses the equivalence of minimizing the smoothed spectral risk and the original spectral risk when losses at the optimal point are different from each other, we find that LSVRG and Prospect often fail to reach the true optimal point, indicating limitations of these methods. In contrast, SOREL converges to the true optimal point in all settings (suboptimality less than 0 means the solution's accuracy is higher than L-BFGS).

## 5.2 FAIR MACHINE LEARNING

In this experiment, we explore the role of the spectral risks in enhancing fairness in machine learning, as studied in Williamson & Menon (2019). We use the `law` and `acs` datasets. `law` refers to the **Law School Admissions Council's National Longitudinal Bar Passage Study**, which is used for the regression task of predicting a student's GPA (Wightman, 1998). `acs` is derived from **US Census surveys**, which is used for the classification task of predicting whether an adult is employed (Ding et al., 2021). All algorithm are implemented using mini-batching in this experiment.

Assume a source distribution $(Y, X, A)$, where $Y$ is the true label, $X$ represents the available features, and $A \in \{0, 1\}$ is the binary sensitive attribute. Let $\hat{Y} = f(X, A)$ be the model's prediction. For binary classification problems, we consider the fairness metric of Equal Opportunity (EO) defined by

$$\text{EO} = \text{P}\{\widehat{Y} = 1 \mid A = 0, Y = 1\} - \text{P}\{\widehat{Y} = 1 \mid A = 1, Y = 1\}.$$

For regression tasks, we consider the absolute mean difference (SMD) as the fairness metric defined by

$$\text{SMD} = \left| \mathbb{E}\left[\hat{Y} \mid A = 1\right] - \mathbb{E}\left[\hat{Y}\right] \right| + \left| \mathbb{E}\left[\hat{Y} \mid A = 0\right] - \mathbb{E}\left[\hat{Y}\right] \right|.$$

Intuitively, if the EO and SMD are close to $0$, the model does not discriminate with respect to $A$. In both datasets, we set race as the sensitive attribute.

**Results.** Tables 2 shows the results of using different spectral risks on the `acs` and `law` datasets, respectively. ERM represents the empirical risk. We find that using spectral risks instead of the empirical risk does improve the fairness metrics of the model, and in most cases, a lower suboptimality indicates better fairness of the model. In the `acs` classification task, SOREL significantly outperforms other algorithms in terms of both fairness metrics and suboptimality. For ESRM, SOREL's suboptimality surpasses that of L-BFGS, while the fairness metric of SGD is worse than the baseline under the ERM setting, possibly due to poor performance of SGD when optimizing the objective function. Additionally, CVaR and Extremile are more effective at reducing EO, compared to Extremile. In the `law` regression task, there is no significant difference in SMD improvement among LSVRG, Prospect, and SOREL, but all perform better than SGD. However, SOREL achieves the lowest suboptimality, and its suboptimality is lower than that of L-BFGS under both the ESRM and Extremile settings. Furthermore, training curves in Appendix B show that SOREL can reach low suboptimality in the shortest amount of time.

Table 2: Results of different algorithms on `acs` and `law`. The values in the ERM row represent the mean fairnes metrics (values closer to $0$ indicate better fairness) on the test set. The first to third rows for each spectral risk (except ERM) represent, respectively: the mean fairness metrics on the test set, relative fairness metric improvements (%) from ERM, and training suboptimality.

| Datasets | acs | | | | law | | | |
|---|---|---|---|---|---|---|---|---|
| ERM | 0.02092 | | | | 0.05188 | | | |
| | SGD | LSVRG | Prospect | SOREL | SGD | LSVRG | Prospect | SOREL |
| CVaR | 0.00645 | 0.00816 | 0.00634 | 0.00551 | 0.04019 | 0.03896 | 0.03893 | 0.03890 |
| | 69.17 | 60.99 | 69.69 | 73.66 | 22.53 | 24.90 | 24.96 | 25.02 |
| | 4.29e-4 | 7.23e-4 | 2.12e-4 | 2.31e-6 | 3.80e-3 | 5.88e-3 | 6.71e-4 | 2.95e-5 |
| ESRM | 0.02469 | 0.01842 | 0.01840 | 0.01770 | 0.04184 | 0.04122 | 0.04123 | 0.04123 |
| | – | 11.95 | 12.05 | 15.39 | 19.35 | 20.55 | 20.53 | 20.53 |
| | 1.47e-3 | 3.33e-4 | 4.38e-6 | -2.38e-8 | 7.60e-4 | 1.13e-4 | 1.52e-7 | -1.13e-7 |
| Extremile | 0.00424 | 0.00377 | 0.00237 | 0.00130 | 0.04416 | 0.04377 | 0.04380 | 0.04381 |
| | 79.73 | 81.98 | 88.67 | 93.97 | 14.88 | 15.63 | 15.57 | 15.56 |
| | 5.11e-3 | 7.90e-3 | 6.69e-4 | 1.14e-4 | 1.63e-4 | 6.14e-4 | 8.14e-6 | -2.12e-7 |

## 5.3 OUT-OF-DISTRIBUTION GENERALIZATION

In this subsection, we explore the role of the spectral risk in enhancing model robustness under distribution shift. We use CVaR and Extremile as the spectral risks. Levy et al. (2020) uses the CVaR measure as the uncertainty set, and their optimization problem is the same as the spectral risk minimization problem (1) that uses CVaR as the spectral risk. We use the `amazon` dataset preprocessed by Mehta et al. (2024b) for the multi-class classification task, which consists of feature representations generated by BERT (Devlin et al., 2019) from the original dataset. `amazon` refers to the **Amazon Reviews** dataset (Ni et al., 2019), which includes textual reviews of products along with their corresponding ratings from one to five, with different reviewers for the training and test sets. We evaluate the worst group classification error (Sagawa et al., 2020) on the test set. Each group is classified based on the true labels. All algorithm are implemented using mini-batching in this experiment.

We also explore the impact of distribution shift on fairness metrics in Section 5.2. In Ding et al. (2021), it is observed that training and testing on different states lead to unpredictable results. We use

data from California as the training data and train models using ERM and CVaR as loss functions, respectively. We then test the models on four other states.

**Results.** Figure 3 shows the results of using CVaR and Extremile spectral risks on `amazon`. SOREL achieves the best worst group classification error in both settings. For CVaR, under similar suboptimality, SOREL reaches the minimum worst group classification error, indicating better generalization performance. Moreover, SOREL is the only algorithm that can converge to the true optimal solution under the CVaR setting. Additionally, SOREL demonstrates optimal or near-optimal convergence rates in both spectral risk settings.

Figure 4 shows the results of models tested on four other states. The circles represent model's performance in California (in-distribution). Models' performance in other states (out-of-distribution) is indeed hard to predict. Notably, models trained with ERM often fail to meet the expected fairness metrics in other states. However, models trained with CVaR often achieves higher test accuracy and better fairness metrics. Moreover, the models trained with SOREL achieve the best or nearly the best EO and test accuracy.

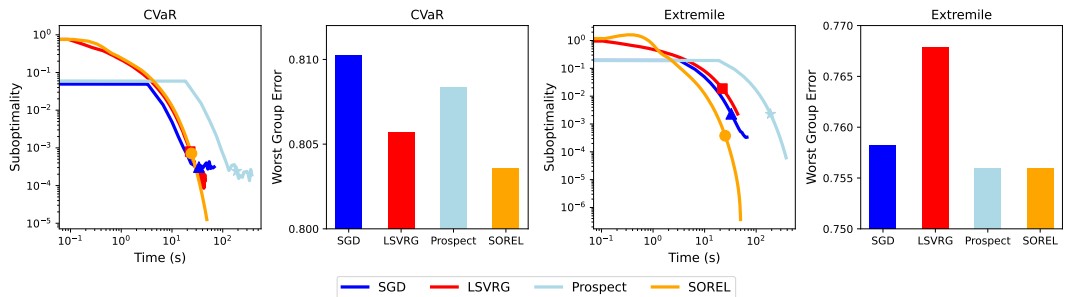

Figure 3: Training curves and worst group classification errors of different algorithms on the `amazon` dataset. The suboptimality at the $500$ th pass (where we evaluate the worst group error) is marked on the training curves. The training curves are extended to illustrate convergence.

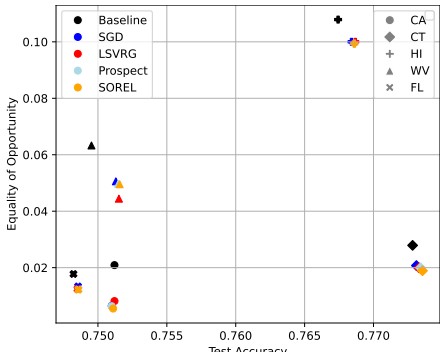

Figure 4: Model performance under geographic distribution shift. The models are trained on the state CA and tested on other states. Dots of different colors (except black) represent the results of using different optimization algorithms to solve the CVaR minimization problem. Baseline refers to the results using ERM as the loss function.

## 6 CONCLUSION

We have proposed SOREL, the first stochastic algorithm with convergence guarantees for the spectral risk minimization problems. We have proved that SOREL achieves a near-optimal rate of $\widetilde{O}(1/\sqrt{\epsilon})$. In experiments, SOREL outperforms existing baselines in terms of sample complexity and runtime in most cases.

Future work includes exploring convergence of SOREL for nonconvex problems, and investigating broader applications of the spectral risk in areas such as fairness and distributionally robust optimization.

## 7 ACKNOWLEDGEMENTS

This work was partly supported by the National Key R&D Program of China under grant 2023YFA1009300, National Natural Science Foundation of China under grants 12171100 and the Major Program of NFSC (72394360,72394364).

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

# A PROOFS

First, we provide an auxiliary lemma. This is an extension of Boob et al. (2024, Lemma 8).

**Lemma 2** *Let $\bar{x}$ be an $\epsilon$-approximate solution of $\min_x\{g(x) + \frac{\lambda}{2}\|x - \hat{x}\|^2\}$ in expectation, where $g : \mathbb{R}^d \to \mathbb{R}$ is $\mu$-strongly convex, $\mu \geq 0$. Then for any $D>0$*

$$\mathbb{E}\{g(\bar{x}) - g(x)\} \leq \mathbb{E}\left\{\frac{\lambda}{2}\left[\|x - \hat{x}\|^2 - \|x - \bar{x}\|^2 - \|\hat{x} - \bar{x}\|^2\right] - \frac{\mu}{2}\|x - \bar{x}\|^2\right\}$$
$$+ \sqrt{\frac{(\lambda + \mu)\epsilon}{2}}D^{-1}\mathbb{E}\|\bar{x} - x\|^2 + \sqrt{\frac{(\lambda + \mu)\epsilon}{2}}D + \epsilon.$$

*Proof:* Let $x^\star = \arg\min_x\{g(x) + \frac{\lambda}{2}\|x - \hat{x}\|^2\}$. By $(\mu + \lambda)$-strong convexity of $g(\cdot) + \frac{\lambda}{2}\|\cdot - \hat{x}\|^2$ we have

$$g(x) + \frac{\lambda}{2}\|x - \hat{x}\|^2 \geq g(x^\star) + \frac{\lambda}{2}\|x^\star - \hat{x}\|^2 + \frac{\mu + \lambda}{2}\|x - x^\star\|^2,$$

$$g(x^\star) - g(x) \leq \frac{\lambda}{2}\left[\|x - \hat{x}\|^2 - \|x^\star - \hat{x}\|^2 - \|x^\star - x\|^2\right] - \frac{\mu}{2}\|x - x^\star\|^2. \tag{7}$$

By the definition of $\bar{x}$ we have

$$\mathbb{E}\{g(\bar{x}) + \frac{\lambda}{2}\|\bar{x} - \hat{x}\|^2\} \leq g(x^\star) + \frac{\lambda}{2}\|x^\star - \hat{x}\|^2 + \epsilon \tag{8}$$

Combining (7) and (8) gives

$$\mathbb{E}\{g(\bar{x}) - g(x)\} \leq \frac{\lambda}{2}\left[\|x - \hat{x}\|^2 - \|x^\star - x\|^2 - \mathbb{E}\|\hat{x} - \bar{x}\|^2\right] - \frac{\mu}{2}\|x - x^\star\|^2 + \epsilon \tag{9}$$

$$= \mathbb{E}\left\{\frac{\lambda}{2}\left[\|x - \hat{x}\|^2 - \|x - \bar{x}\|^2 - \|\bar{x} - \hat{x}\|^2\right] - \frac{\mu}{2}\|x - \bar{x}\|^2\right.$$
$$\left. + \frac{\lambda + \mu}{2}\left[\|x - \bar{x}\|^2 - \|x - x^\star\|^2\right] + \epsilon\right\}$$

$$\leq \mathbb{E}\left\{\frac{\lambda}{2}\left[\|x - \hat{x}\|^2 - \|x - \bar{x}\|^2 - \|\bar{x} - \hat{x}\|^2\right] - \frac{\mu}{2}\|x - \bar{x}\|^2\right.$$
$$\left. + (\lambda + \mu)\|x - \bar{x}\|\|\bar{x} - x^\star\| + \epsilon\right\}, \tag{10}$$

where the last inequality is due to the fact that $\|a\|^2 - \|b\|^2 \leq -2\langle a, b - a\rangle \leq 2\|a\|\|b - a\|$.

Let $x = \bar{x}$ in (9), and take the expectation with respect to $\bar{x}$. Then we have

$$\frac{\lambda + \mu}{2}\mathbb{E}\|x^\star - \bar{x}\|^2 \leq \epsilon.$$

By Hölder's inequality we have

$$\mathbb{E}\|x - \bar{x}\|\|\bar{x} - x^\star\| \leq \left(\mathbb{E}\|x - \bar{x}\|^2\right)^{\frac{1}{2}}\left(\mathbb{E}\|x^\star - \bar{x}\|^2\right)^{\frac{1}{2}}$$

$$\leq \frac{1}{2}\left(\mathbb{E}\|x^\star - \bar{x}\|^2\right)^{\frac{1}{2}}\left(D + D^{-1}\mathbb{E}\|x - \bar{x}\|^2\right).$$

$$\leq \frac{1}{2}\sqrt{\frac{2\epsilon}{\lambda + \mu}}\left(D + D^{-1}\mathbb{E}\|x - \bar{x}\|^2\right).$$

Combining the above results and (10) we get the desired result. $\qquad\square$

Consider solving the problem from Line 8 to Line 16 in Algorithm 1 while updating $w$:

$$\min_w P_k(w) := g(w) + \lambda_{k+1}^\top \ell(w) + \frac{1}{2\eta_k}\|w - w_k\|^2.$$

The following lemma provides the error between $w_{k+1}$ and $\arg\min_w P_k(w)$.

**Lemma 3** *Let $P_k(\boldsymbol{w}) := \boldsymbol{\lambda}_{k+1}^\top \boldsymbol{\ell}(\boldsymbol{w}) + g(\boldsymbol{w}) + \frac{1}{2\tau_k}\|\boldsymbol{w} - \boldsymbol{w}_k\|^2$. Set $\alpha < \frac{L}{4}$, $m_k = \Theta(\frac{L\tau_k}{\mu\tau_k+1})$ and $T_k = O(m_k \log \frac{1}{\epsilon})$ in Algorithm 1. The overall sample complexity of obtaining an $\epsilon$-approximate solution such that $\mathbb{E}P_k(\boldsymbol{w}^{k+1}) - \min_{\boldsymbol{w}} P_k(\boldsymbol{w}) \le \epsilon$ is $O\left(\left(n + \frac{L\tau_k}{\mu\tau_k+1}\right)\log\frac{1}{\epsilon}\right)$. Moreover, we can set $\alpha = \frac{1}{12L}$ and $m_k = \frac{96L}{\mu+\tau_k^{-1}} + 2$ in practice.*

*Proof:* First note that $\boldsymbol{\lambda}^{k+1\top}\boldsymbol{\ell}(\boldsymbol{w})$ is $L$-smooth since

$$\|\sum_{i=i}^n \lambda_{k+1,i}\ell_i(\boldsymbol{x}) - \sum_{i=1}^n \lambda_{k+1,i}\ell_i(\boldsymbol{y})\| \le \sum_{i=1}^n \lambda_{k+1,i}\|\ell_i(\boldsymbol{x}) - \ell_i(\boldsymbol{y})\| \le L\|\boldsymbol{x} - \boldsymbol{y}\|,$$

for $\forall \boldsymbol{x}, \boldsymbol{y} \in \mathbb{R}^d$. In the last inequality we use $\sum_i \lambda_{k+1,i} \le 1$ due to $\boldsymbol{\lambda}_{k+1} \in \Pi_{\boldsymbol{\sigma}}$ and $L$-smoothness of $\ell_i$. Moreover, it is not hard to see that $P_k(\boldsymbol{w})$ is $\mu + \tau_k^{-1}$-strongly convex. By Xiao & Zhang (2014, Theorem 1) we get the desired result. $\qquad\square$

## A.1 PROOF OF LEMMA 1

*Proof:* From the update of $\boldsymbol{\lambda}_{k+1}$ and Lemma 2 we have

$$0 \le \frac{1}{2\eta_k}\left[\|\boldsymbol{\lambda} - \boldsymbol{\lambda}_k\|^2 - \|\boldsymbol{\lambda} - \boldsymbol{\lambda}_{k+1}\|^2 - \|\boldsymbol{\lambda}_{k+1} - \boldsymbol{\lambda}_k\|^2\right] + \langle \boldsymbol{v}_k, \boldsymbol{\lambda}_{k+1} - \boldsymbol{\lambda}\rangle. \qquad (11)$$

From the update of $\boldsymbol{w}_{k+1}$ and Lemma 2 we have

$$\mathbb{E}_k\left\{g(\boldsymbol{w}_{k+1}) + \langle \boldsymbol{\lambda}_{k+1}, \boldsymbol{\ell}(\boldsymbol{w}_{k+1})\rangle - g(\boldsymbol{w}) - \langle \boldsymbol{\lambda}_{k+1}, \boldsymbol{\ell}(\boldsymbol{w})\rangle\right\}$$
$$\le \mathbb{E}_k\left\{\frac{1}{2\tau_k}\left[\|\boldsymbol{w} - \boldsymbol{w}_k\|^2 - \|\boldsymbol{w} - \boldsymbol{w}_{k+1}\|^2 - \|\boldsymbol{w}_{k+1} - \boldsymbol{w}_k\|^2\right] - \frac{\mu}{2}\|\boldsymbol{w} - \boldsymbol{w}_{k+1}\|^2\right.$$
$$\left. + \delta_k + \sqrt{\frac{(\tau_k^{-1} + \mu)\delta_k}{2}}(D^{-1}\|\boldsymbol{w} - \boldsymbol{w}_{k+1}\|^2 + D)\right\}. \qquad (12)$$

Taking the conditional expectation $\mathbb{E}_k$ of both sides of (11) and summing with (12) we obtain that

$$\mathbb{E}_k\left\{L(\boldsymbol{w}_{k+1}, \boldsymbol{\lambda}) - L(\boldsymbol{w}, \boldsymbol{\lambda}_{k+1})\right\}$$
$$= \mathbb{E}_k\left\{g(\boldsymbol{w}_{k+1}) + \langle \boldsymbol{\lambda}, \boldsymbol{\ell}(\boldsymbol{w}_{k+1})\rangle - g(\boldsymbol{w}) - \langle \boldsymbol{\lambda}_{k+1}, \boldsymbol{\ell}(\boldsymbol{w})\rangle\right\}$$
$$\le \mathbb{E}_k\left\{\langle \boldsymbol{\lambda} - \boldsymbol{\lambda}_{k+1}, \boldsymbol{\ell}(\boldsymbol{w}_{k+1})\rangle + \frac{1}{2\eta_k}\left[\|\boldsymbol{\lambda} - \boldsymbol{\lambda}_k\|^2 - \|\boldsymbol{\lambda} - \boldsymbol{\lambda}_{k+1}\|^2 - \|\boldsymbol{\lambda}_{k+1} - \boldsymbol{\lambda}_k\|^2\right]\right.$$
$$+ \frac{1}{2\tau_k}\left[\|\boldsymbol{w} - \boldsymbol{w}_k\|^2 - \|\boldsymbol{w} - \boldsymbol{w}_{k+1}\|^2 - \|\boldsymbol{w}_{k+1} - \boldsymbol{w}_k\|^2\right] + \langle \boldsymbol{v}_k, \boldsymbol{\lambda}_{k+1} - \boldsymbol{\lambda}\rangle \qquad (13)$$
$$\left. - \frac{\mu}{2}\|\boldsymbol{w} - \boldsymbol{w}_{k+1}\|^2 + \delta_k + \sqrt{\frac{(\tau_k^{-1} + \mu)\delta_k}{2}}(D^{-1}\|\boldsymbol{w} - \boldsymbol{w}_{k+1}\|^2 + D)\right\}.$$

$\qquad\square$

## A.2  PROOF OF THEOREM 1

**Lemma 4** *Under the same assumptions as Lemma 1, for any $\boldsymbol{w} \in \mathbb{R}^d$ and $\boldsymbol{\lambda} \in \Pi_{\boldsymbol{\sigma}}$, we have*

$$
\begin{aligned}
&\mathbb{E}_k \left\{ L(\boldsymbol{w}_{k+1}, \boldsymbol{\lambda}) - L(\boldsymbol{w}, \boldsymbol{\lambda}_{k+1}) \right\} \\
\leq &\mathbb{E}_k \left\{ \frac{1}{2\eta_k} \left[ \|\boldsymbol{\lambda} - \boldsymbol{\lambda}_k\|^2 - \|\boldsymbol{\lambda} - \boldsymbol{\lambda}_{k+1}\|^2 \right] + \frac{1}{2\tau_k} \|\boldsymbol{w} - \boldsymbol{w}_k\|^2 - \frac{1}{2} \left( \frac{1}{\tau_k} + \mu \right) \|\boldsymbol{w} - \boldsymbol{w}_{k+1}\|^2 \right. \\
&+ \langle \boldsymbol{\ell}(\boldsymbol{w}_{k+1}) - \boldsymbol{\ell}(\boldsymbol{w}_k), \boldsymbol{\lambda} - \boldsymbol{\lambda}_{k+1} \rangle - \theta_k \langle \boldsymbol{\ell}(\boldsymbol{w}_k) - \boldsymbol{\ell}(\boldsymbol{w}_{k-1}), \boldsymbol{\lambda} - \boldsymbol{\lambda}_k \rangle \\
&- \frac{1}{2} \left[ \frac{1}{\eta_k} - \theta_k \frac{\sqrt{n}G}{\alpha_k} \right] \|\boldsymbol{\lambda}_k - \boldsymbol{\lambda}_{k+1}\|^2 - \frac{1}{2\tau_k} \|\boldsymbol{w}_k - \boldsymbol{w}_{k+1}\|^2 + \frac{\sqrt{n}G\theta_k\alpha_k}{2} \|\boldsymbol{w}_k - \boldsymbol{w}_{k-1}\|^2 \\
&\left. + \delta_k + \sqrt{\frac{(\tau_k^{-1} + \mu)\delta_k}{2}} (D^{-1} \|\boldsymbol{w} - \boldsymbol{w}_{k+1}\|^2 + D) \right\}.
\end{aligned}
$$

(14)

*Proof:*   First, we have

$$
\begin{aligned}
&\langle \boldsymbol{v}_k, \boldsymbol{\lambda}_{k+1} - \boldsymbol{\lambda} \rangle \\
= &\langle \boldsymbol{\ell}(\boldsymbol{w}_k) + \theta_k \left( \boldsymbol{\ell}(\boldsymbol{w}_k) - \boldsymbol{\ell}(\boldsymbol{w}_{k-1}) \right), \boldsymbol{\lambda}_{k+1} - \boldsymbol{\lambda} \rangle \\
= &- \langle \boldsymbol{\ell}(\boldsymbol{w}_{k+1}), \boldsymbol{\lambda} - \boldsymbol{\lambda}_{k+1} \rangle + \langle \boldsymbol{\ell}(\boldsymbol{w}_{k+1}) - \boldsymbol{\ell}(\boldsymbol{w}_k), \boldsymbol{\lambda} - \boldsymbol{\lambda}_{k+1} \rangle \\
&- \theta_k \langle \boldsymbol{\ell}(\boldsymbol{w}_k) - \boldsymbol{\ell}(\boldsymbol{w}_{k-1}), \boldsymbol{\lambda} - \boldsymbol{\lambda}_k \rangle - \theta_k \langle \boldsymbol{\ell}(\boldsymbol{w}_k) - \boldsymbol{\ell}(\boldsymbol{w}_{k-1}), \boldsymbol{\lambda}_k - \boldsymbol{\lambda}_{k+1} \rangle.
\end{aligned}
$$

Then we obtain that

$$
\begin{aligned}
&\langle \boldsymbol{\lambda} - \boldsymbol{\lambda}_{k+1}, \boldsymbol{\ell}(\boldsymbol{w}_{k+1}) \rangle + \langle \boldsymbol{v}_k, \boldsymbol{\lambda}_{k+1} - \boldsymbol{\lambda} \rangle \\
&\leq \langle \boldsymbol{\ell}(\boldsymbol{w}_{k+1}) - \boldsymbol{\ell}(\boldsymbol{w}_k), \boldsymbol{\lambda} - \boldsymbol{\lambda}_{k+1} \rangle - \theta_k \langle \boldsymbol{\ell}(\boldsymbol{w}_k) - \boldsymbol{\ell}(\boldsymbol{w}_{k-1}), \boldsymbol{\lambda} - \boldsymbol{\lambda}_k \rangle \\
&\quad - \theta_k \langle \boldsymbol{\ell}(\boldsymbol{w}_k) - \boldsymbol{\ell}(\boldsymbol{w}_{k-1}), \boldsymbol{\lambda}_k - \boldsymbol{\lambda}_{k+1} \rangle.
\end{aligned}
$$

(15)

Next we bound the last term on the right-hand side of (15):

$$
\begin{aligned}
&\langle \boldsymbol{\ell}(\boldsymbol{w}_k) - \boldsymbol{\ell}(\boldsymbol{w}_{k-1}), \boldsymbol{\lambda}_k - \boldsymbol{\lambda}_{k+1} \rangle \\
\leq &\sqrt{n}G \|\boldsymbol{w}_k - \boldsymbol{w}_{k-1}\| \|\boldsymbol{\lambda}_k - \boldsymbol{\lambda}_{k+1}\| \\
\leq &\frac{\sqrt{n}G\alpha_k}{2} \|\boldsymbol{w}_k - \boldsymbol{w}_{k-1}\|^2 + \frac{\sqrt{n}G}{2\alpha_k} \|\boldsymbol{\lambda}_k - \boldsymbol{\lambda}_{k+1}\|^2,
\end{aligned}
$$

(16)

where the first inequality is due to the $G$-Lipschitz continuity of $\ell_i$ and in the second inequality we use Young's inequality with $\alpha_k > 0$.

Combing (15) and (16) we obtain that

$$
\begin{aligned}
&\langle \boldsymbol{\lambda} - \boldsymbol{\lambda}_{k+1}, \boldsymbol{\ell}(\boldsymbol{w}_{k+1}) \rangle + \langle \boldsymbol{v}_k, \boldsymbol{\lambda}_{k+1} - \boldsymbol{\lambda} \rangle \\
&\leq \langle \boldsymbol{\ell}(\boldsymbol{w}_{k+1}) - \boldsymbol{\ell}(\boldsymbol{w}_k), \boldsymbol{\lambda} - \boldsymbol{\lambda}_{k+1} \rangle - \theta_k \langle \boldsymbol{\ell}(\boldsymbol{w}_k) - \boldsymbol{\ell}(\boldsymbol{w}_{k-1}), \boldsymbol{\lambda} - \boldsymbol{\lambda}_k \rangle \\
&\quad + \frac{\sqrt{n}G\alpha_k\theta_k}{2} \|\boldsymbol{w}_k - \boldsymbol{w}_{k-1}\|^2 + \frac{\sqrt{n}G\theta_k}{2\alpha_k} \|\boldsymbol{\lambda}_k - \boldsymbol{\lambda}_{k+1}\|^2.
\end{aligned}
$$

(17)

Taking the conditional expectation $\mathbb{E}_k$ of both sides of (17) and combing it with Lemma 1 we get the desired result. $\qquad\square$

We remark that $\alpha_k$ does not need to be computed in the actual algorithm but only exists in the theoretical analysis. Next, we try to telescope the terms on the right hand side of (14) by multiplying each term by $\gamma_k$. To ensure that the adjacent terms in the sequence $k = 0, \ldots, K-1$ can be canceled out during summation, we need the parameters of the algorithm to satisfy the following conditions.

**Condition 1** *For $k = 0, 1, ...,$ the following conditions for parameters in the analysis and Algorithm 1:*

$$\frac{\gamma_{k+1}}{\eta_{k+1}} \leq \frac{\gamma_k}{\eta_k}, \tag{18a}$$

$$\frac{\gamma_{k+1}}{\tau_{k+1}} \leq \gamma_k \left( \frac{1}{\tau_k} + \mu - \sqrt{2(\mu + \tau_k^{-1})\delta_k}D^{-1} \right), \tag{18b}$$

$$\gamma_k = \gamma_{k+1}\theta_{k+1}, \tag{18c}$$

$$\sqrt{n}G\alpha_{k+1} \leq \frac{1}{\tau_k}, \tag{18d}$$

$$\theta_k \frac{\sqrt{n}G}{\alpha_k} \leq \frac{1}{\eta_k}. \tag{18e}$$

**Lemma 5** *Assume Assumption 1 holds and Condition 1 is satisfied. Then for all $\boldsymbol{w} \in \mathbb{R}^d$ and $\boldsymbol{\lambda} \in \Pi_{\boldsymbol{\sigma}}$ we have*

$$\frac{\gamma_K}{2\tau_K}\mathbb{E}\|\boldsymbol{w}^\star - \boldsymbol{w}_K\|^2 \leq \frac{\gamma_0}{2\eta_0}\|\boldsymbol{\lambda}^\star - \boldsymbol{\lambda}_0\|^2 + \frac{\gamma_0}{2\tau_0}\|\boldsymbol{w}^\star - \boldsymbol{w}_0\|^2 + \sum_{k=0}^{K-1} \left( \delta_k\gamma_k + \frac{\gamma_k}{2}\sqrt{2(\mu + \tau_k^{-1})\delta_k}D \right),$$

*where $\boldsymbol{w}^\star = \arg\min_{\boldsymbol{w}} R_{\boldsymbol{\sigma}}(\boldsymbol{w}) + g(\boldsymbol{w})$ and $\boldsymbol{\lambda}^\star = \boldsymbol{\sigma}_{\pi^{-1}}$. Here, $\pi$ is the permutation that arranges $\ell_1(\boldsymbol{w}^\star), \ldots, \ell_n(\boldsymbol{w}^\star)$ in ascending order, that is, $\ell_{\pi(1)}(\boldsymbol{w}^\star) \leq \cdots \leq \ell_{\pi(n)}(\boldsymbol{w}^\star)$.*

*Proof:* Taking expectations with respect to $\boldsymbol{w}_k, \ldots, \boldsymbol{w}_1$ in (14) and using the law of total expectation yields

$$\begin{aligned}
&\mathbb{E}\left\{ L(\boldsymbol{w}_{k+1}, \boldsymbol{\lambda}) - L(\boldsymbol{w}, \boldsymbol{\lambda}_{k+1}) \right\} \\
&\leq \mathbb{E}\left\{ \frac{1}{2\eta_k}\left[ \|\boldsymbol{\lambda} - \boldsymbol{\lambda}_k\|^2 - \|\boldsymbol{\lambda} - \boldsymbol{\lambda}_{k+1}\|^2 \right] - \frac{1}{2}\left[ \frac{1}{\eta_k} - \frac{\sqrt{n}G\theta_k}{\alpha_k} \right] \|\boldsymbol{\lambda}_k - \boldsymbol{\lambda}_{k+1}\|^2 \right. \\
&\quad + \langle \boldsymbol{\ell}(\boldsymbol{w}_{k+1}) - \boldsymbol{\ell}(\boldsymbol{w}_k), \boldsymbol{\lambda} - \boldsymbol{\lambda}_{k+1} \rangle - \theta_k\langle \boldsymbol{\ell}(\boldsymbol{w}_k) - \boldsymbol{\ell}(\boldsymbol{w}_{k-1}), \boldsymbol{\lambda} - \boldsymbol{\lambda}_k \rangle \\
&\quad + \frac{1}{2\tau_k}\|\boldsymbol{w} - \boldsymbol{w}_k\|^2 - \frac{1}{2}\left( \frac{1}{\tau_k} + \mu - \sqrt{2(\mu + \tau_k^{-1})\delta_k}D^{-1} \right)\|\boldsymbol{w} - \boldsymbol{w}_{k+1}\|^2 \\
&\quad \left. - \frac{1}{2\tau_k}\|\boldsymbol{w}_k - \boldsymbol{w}_{k+1}\|^2 + \frac{\sqrt{n}G\theta_k\alpha_k}{2}\|\boldsymbol{w}_k - \boldsymbol{w}_{k-1}\|^2 + \delta_k + \frac{1}{2}\sqrt{2(\mu + \tau_k^{-1})\delta_k}D \right\}.
\end{aligned} \tag{19}$$

Multiplying both sides of (19) by $\gamma_k$ and summing over $k = 0$ to $K - 1$ we obtain that

$$\sum_{k=0}^{K-1} \gamma_k \mathbb{E}\left\{ L(\boldsymbol{w}_{k+1}, \boldsymbol{\lambda}) - L(\boldsymbol{w}, \boldsymbol{\lambda}_{k+1}) \right\}$$

$$
\leq \mathbb{E}\left\{ \vphantom{\sum_{k=0}^{K-2}} \right.
\frac{\gamma_0}{2\eta_0} \|\boldsymbol{\lambda} - \boldsymbol{\lambda}_0\|^2 + \sum_{k=0}^{K-2} \underbrace{\frac{1}{2}\left( \frac{\gamma_{k+1}}{\eta_{k+1}} - \frac{\gamma_k}{\eta_k} \right) \|\boldsymbol{\lambda} - \boldsymbol{\lambda}_{k+1}\|^2}_{A} - \frac{\gamma_{K-1}}{2\eta_{K-1}} \|\boldsymbol{\lambda} - \boldsymbol{\lambda}_K\|^2
$$

$$
+ \frac{\gamma_0}{2\tau_0} \|\boldsymbol{w} - \boldsymbol{w}_0\|^2 + \sum_{k=0}^{K-2} \underbrace{\frac{1}{2}\left[ \frac{\gamma_{k+1}}{\tau_{k+1}} - \gamma_k \left( \frac{1}{\tau_k} + \mu - \sqrt{2(\mu + \tau_k^{-1})\delta_k D^{-1}} \right) \right] \|\boldsymbol{w} - \boldsymbol{w}_{k+1}\|^2}_{B}
$$

$$
- \frac{\gamma_{K-1}}{2}\left( \frac{1}{\tau_{K-1}} + \mu - \sqrt{2(\mu + \tau_{K-1}^{-1})\delta_{K-1} D^{-1}} \right) \|\boldsymbol{w} - \boldsymbol{w}_K\|^2
$$

$$
+ \sum_{k=0}^{K-2} \underbrace{(\gamma_k - \gamma_{k+1}\theta_{k+1})}_{C} \langle \boldsymbol{\ell}(\boldsymbol{w}_{k+1}) - \boldsymbol{\ell}(\boldsymbol{w}_k), \boldsymbol{\lambda} - \boldsymbol{\lambda}_{k+1} \rangle + \gamma_{K-1} \langle \boldsymbol{\ell}(\boldsymbol{w}_K) - \boldsymbol{\ell}(\boldsymbol{w}_{K-1}), \boldsymbol{\lambda} - \boldsymbol{\lambda}_K \rangle
$$

$$
+ \frac{1}{2}\sum_{k=0}^{K-2} \underbrace{\left( \gamma_{k+1}\theta_{k+1}\alpha_{k+1}\sqrt{n}G - \frac{\gamma_k}{\tau_k} \right) \|\boldsymbol{w}_k - \boldsymbol{w}_{k+1}\|^2}_{D} - \frac{\gamma_{K-1}}{2\tau_{K-1}} \|\boldsymbol{w}_K - \boldsymbol{w}_{K-1}\|^2
$$

$$
\left. + \frac{1}{2}\sum_{k=0}^{K-1} \underbrace{\left[ -\gamma_k \left( \frac{1}{\eta_k} - \theta_k \frac{\sqrt{n}G}{\alpha_k} \right) \right] \|\boldsymbol{\lambda}_k - \boldsymbol{\lambda}_{k+1}\|^2}_{E} + \sum_{k=0}^{K-1}\left( \delta_k \gamma_k + \frac{\gamma_k}{2}\sqrt{2(\mu + \tau_k^{-1})\delta_k D} \right) \right\}.
$$

Here we use $\boldsymbol{\ell}(\boldsymbol{w}_0) - \boldsymbol{\ell}(\boldsymbol{w}_{-1}) = 0$ by $\boldsymbol{w}_0 = \boldsymbol{w}_{-1}$ and $\boldsymbol{\lambda}_0 = \boldsymbol{\lambda}_{-1}$. By Condition 1, we have $A, B, D, E \leq 0$ and $C = 0$.

Then we have

$$\sum_{k=0}^{K-1} \gamma_k \mathbb{E}\left\{ L(\boldsymbol{w}_{k+1}, \boldsymbol{\lambda}) - L(\boldsymbol{w}, \boldsymbol{\lambda}_{k+1}) \right\}$$

$$
\leq \mathbb{E}\left\{ \frac{\gamma_0}{2\eta_0} \|\boldsymbol{\lambda} - \boldsymbol{\lambda}_0\|^2 - \frac{\gamma_{K-1}}{2\eta_{K-1}} \|\boldsymbol{\lambda} - \boldsymbol{\lambda}_K\|^2 + \frac{\gamma_0}{2\tau_0} \|\boldsymbol{w} - \boldsymbol{w}_0\|^2 \right.
$$

$$
- \frac{\gamma_{K-1}}{2}\left( \frac{1}{\tau_{K-1}} + \mu - \sqrt{2(\mu + \tau_{K-1}^{-1})\delta_{K-1} D^{-1}} \right) \|\boldsymbol{w} - \boldsymbol{w}_K\|^2 \tag{20}
$$

$$
+ \gamma_{K-1} \langle \boldsymbol{\ell}(\boldsymbol{w}_K) - \boldsymbol{\ell}(\boldsymbol{w}_{K-1}), \boldsymbol{\lambda} - \boldsymbol{\lambda}_K \rangle - \frac{\gamma_{K-1}}{2\tau_{K-1}} \|\boldsymbol{w}_K - \boldsymbol{w}_{K-1}\|^2
$$

$$
\left. + \sum_{k=0}^{K-1}\left( \delta_k \gamma_k + \frac{\gamma_k}{2}\sqrt{2(\mu + \tau_k^{-1})\delta_k D} \right) \right\}.
$$

Next we bound $\gamma_{K-1}\langle \boldsymbol{\ell}(\boldsymbol{w}_K) - \boldsymbol{\ell}(\boldsymbol{w}_{K-1}), \boldsymbol{\lambda} - \boldsymbol{\lambda}_K \rangle$ similar to (16). We have

$$\langle \boldsymbol{\ell}(\boldsymbol{w}_K) - \boldsymbol{\ell}(\boldsymbol{w}_{K-1}), \boldsymbol{\lambda} - \boldsymbol{\lambda}_K \rangle \leq \frac{\sqrt{n}G\alpha_K}{2} \|\boldsymbol{w}_K - \boldsymbol{w}_{K-1}\|^2 + \frac{1}{2}\frac{\sqrt{n}G}{\alpha_K} \|\boldsymbol{\lambda} - \boldsymbol{\lambda}_K\|^2.$$

Taking the expectation and plugging this into (20), we obtain that

$$\sum_{k=0}^{K-1} \gamma_k \mathbb{E}\left\{L(\boldsymbol{w}_{k+1}, \boldsymbol{\lambda}) - L(\boldsymbol{w}, \boldsymbol{\lambda}_{k+1})\right\}$$

$$\leq \mathbb{E}\left\{ \frac{\gamma_0}{2\eta_0}\|\boldsymbol{\lambda} - \boldsymbol{\lambda}_0\|^2 - \frac{1}{2}\underbrace{\left[\frac{\gamma_{K-1}}{\eta_{K-1}} - \gamma_{K-1}\frac{\sqrt{n}G}{\alpha_K}\right]}_{\tilde{A}}\|\boldsymbol{\lambda} - \boldsymbol{\lambda}_K\|^2 \right.$$

$$+ \frac{\gamma_0}{2\tau_0}\|\boldsymbol{w} - \boldsymbol{w}_0\|^2 - \underbrace{\frac{\gamma_{K-1}}{2}\left(\frac{1}{\tau_{K-1}} + \mu - \sqrt{2(\mu + \tau_{K-1}^{-1})\delta_{K-1}D^{-1}}\right)}_{\tilde{B}}\|\boldsymbol{w} - \boldsymbol{w}_K\|^2$$

$$\left. + \frac{\gamma_{K-1}}{2}\underbrace{\left(\alpha_K\sqrt{n}G - \frac{1}{\tau_{K-1}}\right)}_{\tilde{C}}\|\boldsymbol{w}_K - \boldsymbol{w}_{K-1}\|^2 + \sum_{k=0}^{K-1}\left(\delta_k\gamma_k + \frac{\gamma_k}{2}\sqrt{2(\mu + \tau_k^{-1})\delta_k}D\right)\right\}.$$

$$(21)$$

We analyze $\tilde{A}$-$\tilde{D}$ under Condition 1:

$$\tilde{A} \overset{(18a)}{\geq} \left[\frac{\gamma_K}{\eta_K} - \gamma_{K-1}\frac{\sqrt{n}G}{\alpha_K}\right] \overset{(18c)}{=} \gamma_K\left[\frac{1}{\eta_K} - \theta_K\frac{\sqrt{n}G}{\alpha_K}\right] \overset{(18e)}{\geq} 0,$$

$$\tilde{B} \overset{(18b)}{\geq} \frac{\gamma_K}{2\tau_K},$$

$$\tilde{C} \overset{(18d)}{\leq} 0.$$

We obtain that

$$\sum_{k=0}^{K-1} \gamma_k \mathbb{E}\left\{L(\boldsymbol{w}_{k+1}, \boldsymbol{\lambda}) - L(\boldsymbol{w}, \boldsymbol{\lambda}_{k+1})\right\}$$

$$\leq \frac{\gamma_0}{2\eta_0}\|\boldsymbol{\lambda} - \boldsymbol{\lambda}_0\|^2 + \frac{\gamma_0}{2\tau_0}\|\boldsymbol{w} - \boldsymbol{w}_0\|^2 - \frac{\gamma_K}{2\tau_K}\mathbb{E}\|\boldsymbol{w} - \boldsymbol{w}_K\|^2 + \sum_{k=0}^{K-1}\left(\delta_k\gamma_k + \frac{\gamma_k}{2}\sqrt{2(\mu + \tau_k^{-1})\delta_k}D\right).$$

$$(22)$$

For any $\boldsymbol{w} \in \mathbb{R}^d$ and $\boldsymbol{\lambda} \in \Pi_{\boldsymbol{\sigma}}$, we have $L(\boldsymbol{w}^\star, \boldsymbol{\lambda}^\star) = \max_{\boldsymbol{\lambda} \in \Pi_{\boldsymbol{\sigma}}} L(\boldsymbol{w}^\star, \boldsymbol{\lambda}) \geq L(\boldsymbol{w}^\star, \boldsymbol{\lambda})$. On the other hand, we have $L(\boldsymbol{w}, \boldsymbol{\lambda}^\star) \geq L(\boldsymbol{w}^\star, \boldsymbol{\lambda}^\star) = \min_{\boldsymbol{w}} L(\boldsymbol{w}, \boldsymbol{\lambda}^\star)$. Thus we obtain that $L(\boldsymbol{w}_{k+1}, \boldsymbol{\lambda}^\star) - L(\boldsymbol{w}^\star, \boldsymbol{\lambda}_{k+1}) \geq 0$ for $\forall k = 0, \ldots, K-1$. Let $\boldsymbol{w} = \boldsymbol{w}^\star$ and $\boldsymbol{\lambda} = \boldsymbol{\lambda}^\star$ in (22) we get the desired result. □

Now we are ready to prove Theorem 1. By choosing appropriate parameters in Algorithm 1 to satisfy Condition 1, we can achieve the desired convergence rate.

**Proof of Theorem 1.**

*Proof:* First, we obtain an $\delta_k$ approximate solution to (11) through $T_k$ updates to $\boldsymbol{w}$ in Algorithm 1 by Lemma 3. We then verify that Condition 1 is satisfied by the parameters.

It is not hard to see that $\frac{\gamma_{k+1}}{\gamma_k} = \frac{\eta_{k+1}}{\eta_k} = \frac{k+2}{k+1}$ and $\theta_{k+1} = \frac{\gamma_k}{\gamma_{k+1}} = \frac{k+1}{k+2}$. Thus (18a) and (18c) are satisfied.

Since $\delta_k \leq \frac{\mu}{8(k+5)}D^2$, we have $\sqrt{2(\mu + \tau_k^{-1})\delta_k}D^{-1} \leq \sqrt{2\mu(1 + \frac{k+1}{4})\frac{\mu D^2}{8(k+5)}}D^{-1} \leq \frac{\mu}{4}$. Then we obtain that

$$\frac{\gamma_{k+1}}{\gamma_k\tau_{k+1}} = \frac{k+2}{4}\mu + \frac{k+2}{4(k+1)}\mu \leq \frac{k+4}{4}\mu,$$

and

$$\frac{1}{\tau_k} + \mu - \sqrt{2(\mu + \tau_k^{-1})\delta_k}D^{-1} \geq \frac{k+1}{4}\mu + \mu - \frac{\mu}{4} = \frac{k+4}{4}\mu.$$

Thus (18b) holds.

Furthermore, (18d) and (18e) hold due to $\sqrt{n}G\alpha_{k+1} = nG^2\eta_k = \frac{k+1}{8}\mu \leq \frac{k+1}{4}\mu = \frac{1}{\tau_k}$ and $\theta_k \frac{\sqrt{n}G}{\alpha_k} = \frac{\eta_{k-1}}{\eta_k}\frac{\sqrt{n}G}{\sqrt{n}G\eta_{k-1}} = \frac{1}{\eta_k}$.

Now Condition 1 is satisfied. By Lemma 5, we have

$$\frac{\gamma_K}{2\tau_K}\mathbb{E}\|\boldsymbol{w} - \boldsymbol{w}_K\|^2 \leq \frac{\gamma_0}{2\eta_0}\|\boldsymbol{\lambda}^\star - \boldsymbol{\lambda}_0\|^2 + \frac{\gamma_0}{2\tau_0}\|\boldsymbol{w}^\star - \boldsymbol{w}_0\|^2 + \sum_{k=0}^{K-1}\left(\delta_k\gamma_k + \frac{\gamma_k}{2}\sqrt{2(\mu + \tau_k^{-1})\delta_k D}\right).$$

Since $\delta_k \leq \mu(k+1)^{-6}D^2$, we have $\sum_{k=0}^\infty \delta_k\gamma_k \leq \mu D^2\sum_{k=0}^\infty(k+1)^{-5} \leq \frac{\mu}{4}D^2$, and

$$\sum_{k=0}^\infty \gamma_k\sqrt{(\mu + \tau_k^{-1})\delta_k}D \leq \frac{\sqrt{2}\mu}{4}D^2\sum_{k=0}^\infty(k+1)^{-2}\sqrt{k+5} \leq \frac{\sqrt{2}\mu}{4}D^2\sum_{k=0}^\infty(k+1)^{-2}\left(\sqrt{k+1}+2\right) \leq \sqrt{2}\mu D^2.$$

Finally, by $\frac{\gamma_K}{2\tau_K} = \frac{\mu(K+1)^2}{8}, \tau_0 = \frac{4}{\mu}, \eta_0 = \frac{\mu}{8nG^2}$ and $D = \frac{G}{\mu}$, we get the desired result. $\qquad\square$

## A.3 PROOF OF COROLLARY 1

*Proof:* Recall that $\tau_k = \frac{4}{\mu(k+1)}$. It is not hard to see that $\frac{L\tau_k}{\mu\tau_k+1} = \frac{4L}{\mu(k+5)} \leq \frac{4L}{\mu(k+1)}$. By Lemma 3, we get a $\delta_k$ approximate solution with the sample complexity of $C_{\boldsymbol{w}_{k+1}} = O\left(\left(n + \frac{L}{\mu(k+1)}\right)\log\left(\delta_k^{-1}\right)\right)$. We set $\delta_k = D^2\min\left(\frac{\mu}{8(k+5)}, \mu(k+1)^{-6}\right) = \frac{G^2}{\mu}(k+1)^{-6}$ for $k \geq 1$. And $\delta_0 = \mu/40$. The total sample complexity is

$$\sum_{k=0}^{K-1}C_{\boldsymbol{w}_{k+1}} = \sum_{k=0}^{K-1}O\left(\left(n + \frac{L}{\mu(k+1)}\right)\left(\log(k+1) + \log\left(\frac{\mu}{G^2}\right)\right)\right)$$
$$= O\left(nK\log\frac{\mu K}{G^2} + \frac{L}{\mu}\log K\log\frac{\mu K}{G^2}\right).$$

In the last equality, we calculate $\sum_{k=1}^K \frac{\log k}{k} = O\left((\log K)^2\right)$, $\sum_{k=1}^K \log k = O(K\log K)$ and $\sum_{k=1}^K \frac{1}{k} = O(\log K)$. By Theorem 1, to achieve an $\epsilon$-optimal solution, we need $K = O\left(\frac{\sqrt{n}G}{\mu\sqrt{\epsilon}}\right)$. Therefore, the total sample complexity is $O\left(\frac{n^{\frac{3}{2}}G}{\mu\sqrt{\epsilon}}\log\frac{1}{\epsilon} + \frac{L}{\mu}\left(\log\frac{1}{\epsilon}\right)^2\right)$. $\qquad\square$

# B EXPERIMENTAL DETAILS AND ADDITIONAL EXPERIMENTAL RESULTS

We now outline the details of our experimental setup. Our experimental setup mainly follows that of Mehta et al. (2024b).

**Datasets.** We use the same five datasets from the regression task in Section 5.1 and the `amazon` dataset in Section 5.3 as in Mehta et al. (2024b). The statistical characteristics are summarized in Table 3.

Other two datasets in Section 5.2 are as follows:

1. `acs`: predicting whether an American adult is employed.

2. `law`: predicting a student's GPA.

In the experiments, we normalize the features of the sample matrix $\boldsymbol{X} \in \mathbb{R}^{n\times d}$ so that each feature has a mean of $0$ and a variance of $1$. The test sets are normalized using the statistics of the training set.

| Dataset | # features | # samples | Source |
|---------|-----------|-----------|--------|
| yacht | 6 | 244 | Tsanas & Xifara (2012) |
| energy | 8 | 614 | Baressi Šegota et al. (2019) |
| concrete | 8 | 824 | Yeh (2006) |
| kin8nm | 8 | 6,553 | Akujuobi & Zhang (2017) |
| power | 4 | 7,654 | Tüfekci (2014) |
| acs | 16 | 10000 | Ding et al. (2021) |
| law | 10 | 20800 | Wightman (1998) |
| amazon | 535 | 20000 | Mehta et al. (2024b),Ni et al. (2019) |

Table 3: Statistical details of real datasets and sources.

**Objectives.** We use linear models in our experiments. For spectral risks, we adopt three types: ESRM, Extremile, and CVaR, as specified in Table 1. Additionally, we set the regularizer $g(\boldsymbol{w})$ to $\frac{\mu}{2}\|\boldsymbol{w}\|^2$ with $\mu = \frac{1}{n}$, where $n$ donates the number of sample points in the taining set. Thus, Problem (1) can be written as

$$\min_{\boldsymbol{w}} \sum_{i=1}^{n} \sigma_i \ell_{[i]}(\boldsymbol{w}) + \frac{\mu}{2}\|\boldsymbol{w}\|^2,$$

where $\ell_i(\cdot)$ is the loss function, which will be chosen in different forms for different tasks.

**Hyperparameter Selection.** We use similar hyperparameter selection method as in Mehta et al. (2024b). We set the batch size for SGD to $64$. For the selection of step size $\alpha$, we set the random seed $s \in \{1, \ldots, S\}$. For a single seed $s$, we calculate the average training loss of the last ten epochs, donated by $L_s(\alpha)$. We choose $\alpha$ that minimizes $\frac{1}{S}\sum_{s=1}^{S} L_s(\alpha)$, where $\alpha \in \{1 \times 10^{-4}, 3 \times 10^{-4}, 1 \times 10^{-3}, 3 \times 10^{-3}, 1 \times 10^{-2}, 3 \times 10^{-2}, 1 \times 10^{-1}, 3 \times 10^{-1}\}$. For LSVRG, we set the length of an epoch to $n$. For SOREL, we set $T_k = m_k = n$. Moreover, we set batch size to 64 for all algorithms with mini-batching.

For SOREL, we follow the parameter values given in Theorem 1. In particular, we set $\theta_k = \frac{k}{k+1}$ and $\tau_k = \frac{20n}{k+1}$ in all experiments. Therefore, there are only two parameters $\alpha$ and $\eta_k$ left to tune. We set $\eta_k = \frac{C(k+1)}{n}$ and choose $C$ from $\{1 \times 10^{-2}, 2 \times 10^{-2}, 4 \times 10^{-2}, 1 \times 10^{-1}, 2 \times 10^{-1}, 4 \times 10^{-1}, 1 \times 10^{0}, 2 \times 10^{0}, 4 \times 10^{0}, 1 \times 10^{1}\}$, with two orders of magnitude higher numbers used in law, since the Lipschitz constant $G$ is hard to estimate. We use grid search to select $\alpha$ and $C$, with the selection criteria being the same as the previous paragraph. We apply stochastic gradient descent to solve (5) instead of proximal stochastic gradient descent.

**Experimental Environment.** We run all experiments on a laptop with 16.0 GB RAM and Intel i7-1360P 2.20 GHz CPU. All algorithms are implemented in Python 3.8.

### B.1 EXPERIMENTAL DETAILS ON LINEAR REGRESSION

**Dataset.** We use the same dataset as that used in Mehta et al. (2024b), as previously described.

**Objectives.** We use the least square loss in this experiment. For spectral risks, we adopt three types: ESRM ($\rho = 2$), Extremile ($r = 2.5$), and CVaR ($\alpha = 0.5$).

**Evaluation.** We set random seeds $s \in \{1, 2, 3, 4, 5\}$ as the seeds for the random algorithms. We compare the suboptimality versus passes (the number of samples divided by $n$) and runtime. The suboptimality is defined as

$$\text{Suboptimality}(\boldsymbol{w}_k) = \frac{R_{\boldsymbol{\sigma}}(\boldsymbol{w}_k) + g(\boldsymbol{w}_k) - R_{\boldsymbol{\sigma}}(\boldsymbol{w}^\star) - g(\boldsymbol{w}^\star)}{R_{\boldsymbol{\sigma}}(\boldsymbol{w}_0) + g(\boldsymbol{w}_0) - R_{\boldsymbol{\sigma}}(\boldsymbol{w}^\star) - g(\boldsymbol{w}^\star)},$$

where $\boldsymbol{w}^\star$ is calculated by L-BFGS (Nocedal & Wright, 1999).

## B.2 EXPERIMENTAL DETAILS ON FAIR MACHINE LEARNING

**Dataset.** We use two datasets, `acs` (Ding et al., 2021) and `law` (Wightman, 1998), which are for classification and regression tasks, respectively. For `acs`, we randomly selected $10,000$ sample points from the California data. We use data from four states Connecticut, Hawaii, West Virginia and Florida in Ding et al. (2021) as the test dataset to explore the out-of-distribution performance of models trained with spectral risks, with each state having $36287, 14400, 18066$, and $202160$ sample points, respectively.

**Objectives.** For the regression and classification tasks, we use the least squares loss and the binary logistic loss, respectively. For `acs`, we set the spectral risks to CVaR ($\alpha = 0.75$), ESRM ($\rho = 1.75$) and Extremile ($r = 2.1$). For `law`, we set the spectral risks to CVaR ($\alpha = 0.05$), ESRM ($\rho = 20$) and Extremile ($r = 10$).

**Evaluation.** We fix the number of training passes at $100$. We split the training set and test set in a 4:1 ratio and used five-fold cross-validation to report the average results on the test set. For each training and test set split, we set random seeds $s \in \{1, 2, 3\}$ as the seeds for the random algorithms. All algorithms are implemented using mini-batching. We set race as the sensitive feature. For `acs`, the sensitive feature includes Black and White. For `law`, the sensitive feature includes non-White and White. For the task of exploring models' out-of-domain performance, we directly use the models obtained from the `acs` experiments as the models trained on California dataset. Then, we test these models on all data points from the other four states.

## B.3 EXPERIMENTAL DETAILS ON OUT-OF-DISTRIBUTION GENERALIZATION

**Dataset.** `amazon` (Ni et al., 2019) is for the multi-class classification task. We use the preprocessed data in Mehta et al. (2024b). They fine-tuned a BERT model on $10,000$ held-out examples and applied PCA to the deep representations produced by BERT. The training set and test set each contain 10,000 samples. #features in Table 3 refers to the total dimension of the parameter vectors for all 5 classes.

**Objectives.** We use a linear model and the multinomial logistic loss. In `amazon`, we set the spectral risks to CVaR ($\alpha = 0.75$) and Extremile ($r = 2.0$).

**Evaluation.** We set random seeds $s \in \{1, 2, 3, 4, 5\}$ as the seeds for the random algorithms. We evaluate the worst group classification error (Sagawa et al., 2020) on the test set. Each group is classified based on the true labels. We fix the number of passes during training to $500$ and report the average worst group classification error of the last ten passes. All algorithms are implemented using mini-batching.

## B.4 ADDITIONAL EXPERIMENTAL RESULTS

**Algorithms with mini-batching.** In Figure 5, we present results of the algorithms with mini-batching for tasks in Section 5.1. Mini-batching has a significant improvement on the convergence rate of all the algorithms. Similar to what is shown in Figure 2, SGD, LSVEG and Prospect fail to converge to the true optimal points, especially in the first two datasets. SOREL converges to the optimal solutions in all settings, and achieves the best or competitive results, in terms of sample complexity, runtime, or both, except in the setting of CVaR and `power` dataset. Still SOREL performs competitively for the suboptimality of $10^{-7}$ in this setting.

**Training curves in fair machine learning.** Figure 6 shows the training curves for the task in Section 5.2. The training curves are extended to illustrate convergence. SOREL is able to achieve low suboptimality in the shortest amount of time.

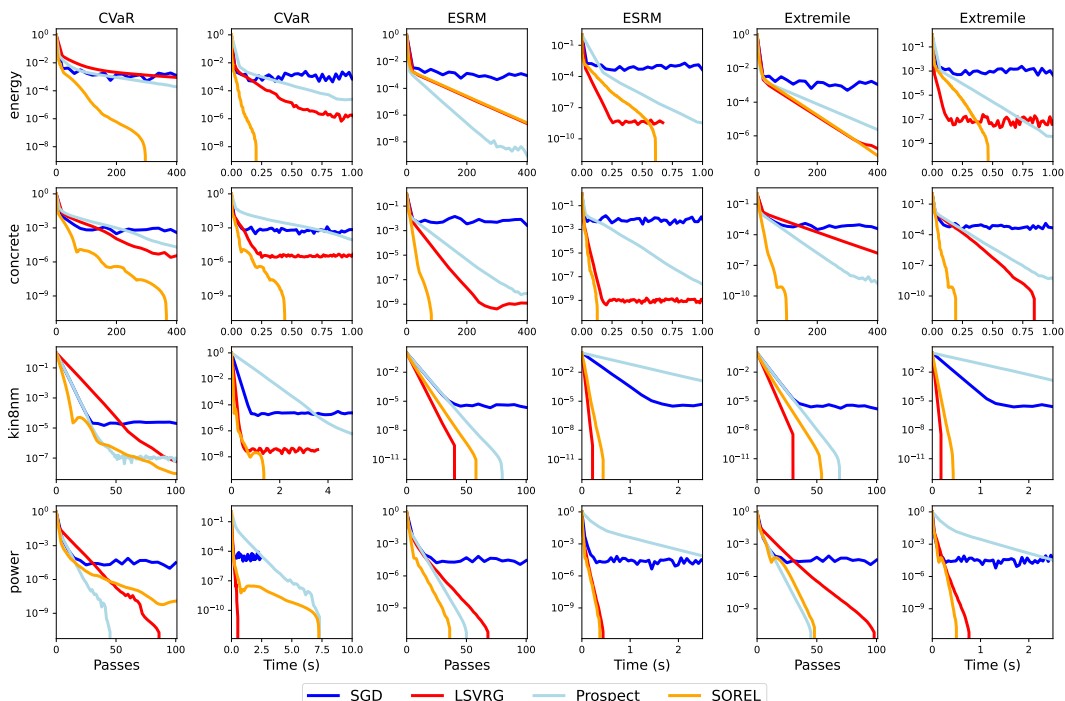

Figure 5: Suboptimality of spectral risks for different algorithms **with mini-batching**. The $x$-axis represents the effective number of samples used by the algorithm divided by $n$ (odd columns) or CPU time (even columns).

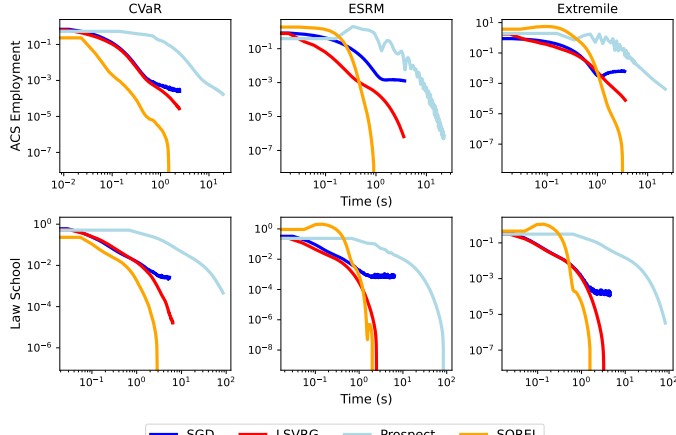

Figure 6: Suboptimality of spectral risks for different algorithms on fairness benchmarks. The $x$-axis represents the CPU time.

## C  EXAMPLE

To illustrate the necessity of stabilizing the trajectory of the primal variable in Section 3.1, we provide a toy example. For simplicity, we consider a one-dimensional problem

$$\min_{w \in \mathbb{R}} \sigma_1 \ell_{[1]}(w) + \sigma_2 \ell_{[2]}(w), \tag{23}$$

where $\sigma_1 = 0, \sigma_2 = 1$ and $\ell_1 = \frac{1}{2}(w-1)^2, \ell_2(w) = \frac{1}{2}(w+1)^2$. We use the following deterministic method, similar to Algorithm 1.

**Example 1** *For any $0 < \alpha < 2$, suppose we solve Problem (23) using Algorithm 2 and $T$ is sufficiently large. In that case, the iterative sequence $\{w_k\}$ can not converge to the optimal solution for any initial point $w_0$.*

---

**Algorithm 2** Simplified Algorithm for Solving the Example Problem.

1: **for** $k = 0, 1, \ldots$ **do**
2:    Update $\{\lambda_{k+1,1}, \lambda_{k+1,2}\} = \{\sigma_1, \sigma_2\}$ if $\ell_1(w_k) \geq \ell_2(w_k)$ else $\{\sigma_2, \sigma_1\}$. Set $w_k^0 = w_k$.
3:    **for** $t = 0, 1, \ldots, T - 1$ **do**
4:       Compute the gradient $g^t = \lambda_{k+1,1}\nabla \ell_1(w_k^t) + \lambda_{k+1,2}\nabla \ell_2(w_k^t)$.
5:       Update $w^{t+1} = w^t - \alpha g^t$.
6:    **end for**
7:    Set $w_{k+1} = w_k^T$.
8: **end for**

---

Without loss of generality, we assume $w_0 > 0$, in which case $\boldsymbol{\lambda}_1 = [0, 1]^\top$. We solve $\min_{w_1 \in \mathbb{R}} \frac{1}{2}(w_1 + 1)^2$ through sufficient steps of gradient descent to obtain $w_1 = -1$. At this point, $\boldsymbol{\lambda}_2 = [1, 0]^\top$. By iterating this process, $w_k$ always oscillates between -1 and 1, unable to converge to $w^\star = 0$. If $w_0 = 0$, we set $\sigma_1 = 1$ and $\sigma_2 = 0$, reaching the same conclusion. A similar conclusion can be extended to stochastic methods in the expectation sense.

We know that $\ell_1(w^\star) = \ell_2(w^\star)$ at the optimal point $w^\star = 0$. Clearly, the iterative sequence of the algorithm oscillates at $w^\star$ and cannot converge to the optimal solution. Although Mehta et al. (2022; 2024b) employ a similar approach to update $\boldsymbol{\lambda}$ for subgradient estimations, they consider the smoothed spectral risk by adding a strongly concave term with respect to $\boldsymbol{\lambda}$. However, for the original spectral risk minimization problems, updating $\boldsymbol{\lambda}$ with their method results in discontinuities, thereby lacking convergence guarantees.

## D   SOREL WITH MINI-BATCHING

In this section, we present the results of SOREL with mini-batching. To apply mini-batching, We only need to change Line 13 of Algorithm 1 to: Sample a mini-batch $b_t \subset \{1, \ldots, n\}$ without replacement, and change Line 14 to

$$\boldsymbol{d}_{k,t} = \frac{1}{b} \sum_{i \in b_t} [n\lambda_{i,k+1}\nabla \ell_i(\boldsymbol{w}_{k,t}) - n\lambda_{i,k+1}\nabla \ell_i(\bar{\boldsymbol{w}})] + \bar{\boldsymbol{g}},$$

where $b = |b_t|$ is the mini-batch size.

We first present the main result of SOREL with mini-batching.

**Corollary 2** *Use the same conditions in Therorem 1. Additionally, set the step-size $\alpha = \frac{b(n-1)}{5L(n-b)}$, $m_k = \frac{400L(n-b)}{(k+5)\mu b(n-1)} + 8$ and $T_k = O(m_k \log \frac{1}{\delta_k})$. Then we obtain an output $\boldsymbol{w}_K$ of SOREL with mini-batching such that $\mathbb{E}\|\boldsymbol{w}_k - \boldsymbol{w}^\star\|^2 \leq \epsilon$ in a total sample complexity of $O\left(\frac{n^{\frac{3}{2}}G}{\mu\sqrt{\epsilon}} \log \frac{1}{\epsilon} + \frac{L(n-b)}{\mu(n-1)} \left(\log \frac{1}{\epsilon}\right)^2\right)$.*

### D.1   PROOF OF COROLLARY 2

We first discuss the inner loop in Lines 6-17 of Algorithm 1. This is an extension of SVRG (Johnson & Zhang, 2013; Xiao & Zhang, 2014). To illustrate more clearly, we consider the problem

$$\min_{\boldsymbol{w}} P(\boldsymbol{w}) := F(\boldsymbol{w}) + h(\boldsymbol{w}),$$

where $F(\boldsymbol{w}) = \frac{1}{n} \sum_{i=1}^n f_i(\boldsymbol{w})$, each $f_i$ is convex and $L$-smooth, and $h$ is $\mu$-strongly convex. We rewrite Lines 6-17 of Algorithm 1 to Algorithm 3. Note that, by setting $\bar{\boldsymbol{w}}_0 = \boldsymbol{w}_k$, $m = m_k$, $F(\boldsymbol{w}) = \boldsymbol{\lambda}_{k+1}^\top \ell(\boldsymbol{w})$, and $h(\boldsymbol{w}) = g(\boldsymbol{w}) + \frac{1}{2\tau_k}\|\boldsymbol{w} - \boldsymbol{w}_k\|^2$, Algorithm 3 is the same as Lines 6-17 of Algorithm 1.

**Assumption 2** *Each $f_i : \mathbb{R}^d \to \mathbb{R}$ is convex and $L$-smooth. $h : \mathbb{R}^d \to \mathbb{R} \cup \{\infty\}$ is proper, lower semicontinuous and $\mu_h$-strongly convex.*

The following two results are adopted from Xiao & Zhang (2014), which will be used in the proof of the main result.

---

**Algorithm 3** Simplified Inner Loop of Algorithm 1 with Mini-batching.

1: **Input:** initial $\bar{w}_0$, the learning rate $\alpha$, mini-batch size $b$, and the inner-loop length $m$.
2: **for** $s = 0, 1, \ldots$ **do**
3:     $\bar{w} = \bar{w}_{s-1}, \bar{g} = \nabla F(\bar{w})$.
4:     $w_0 = \bar{w}$.
5:     **for** $t = 0, 1, \ldots, m-1$ **do**
6:         Sample $b_t \subset \{1, \ldots, n\}$ of size $b$ uniformly at random without replacement.
7:         $d_t = \frac{1}{b} \sum_{i \in b_t} [\nabla f_i(w_t) - \nabla f_i(\bar{w})] + \bar{g}$.
8:         $w_{t+1} = \text{Prox}_{\alpha h}\{w_t - \alpha d_t\}$.
9:     **end for**
10:    $\bar{w}_s = \frac{1}{m} \sum_{t=1}^{m} w_t$.
11: **end for**

---

**Lemma 6** (Xiao & Zhang, 2014, Lemma 1) *Suppose Assumption 2 holds, and let $w_\star = \arg\min_w P(w)$. Then for all $w \in \mathbb{R}^d$*

$$\frac{1}{n}\|\nabla f_i(w) - \nabla f_i(w_\star)\|^2 \le 2L\left(P(w) - P(w_\star)\right).$$

**Lemma 7** (Xiao & Zhang, 2014, Lemma 3) *Suppose Assuption 2 holds, let $\Delta_t = d_t - \nabla F(w_t)$ and $w_\star = \arg\min_w P(w)$. Then*

$$\|w_{t+1} - w_\star\|^2 \le \|w_t - w_\star\|^2 - 2\alpha\left[P\left(w_{t+1}\right) - P\left(w_\star\right)\right] - 2\alpha\Delta_t^\top\left(w_{t+1} - w_\star\right). \tag{24}$$

The following lemma bounds the variance of the stochastic gradient $d_t$.

**Lemma 8** *Let $\mathbb{E}_t$ be the conditional expectation given $w_t$ and $w_\star = \arg\min_w P(w)$. We have*

$$\mathbb{E}_t\|d_t - \nabla F(w_t)\|^2 \le \frac{2(n-b)L}{b(n-1)}\left(P(w_t) - P(w_\star) + P(\bar{w}) - P(w_\star)\right).$$

*Proof:* Define $\xi_i = \nabla f_i(w_t) - \nabla f_i(\bar{w})$.

$$\mathbb{E}_t\|d_t - \nabla F(w_t)\|^2$$

$$= \mathbb{E}_t\|\frac{1}{b}\sum_{i_t \in b_t}\left(\nabla f_{i_t}(w_t) - \nabla f_{i_t}(\bar{w})\right) + \frac{1}{n}\sum_{i=1}^{n}\left(\nabla f_i(\bar{w}) - \nabla f_i(w_t)\right)\|^2$$

$$= \mathbb{E}_t\|\frac{1}{b}\sum_{i_t \in b_t}\xi_{i_t}\|^2 - \|\frac{1}{n}\sum_{i=1}^{n}\xi_i\|^2$$

$$= \frac{1}{b^2}\mathbb{E}_t\left[\sum_{i_t \neq j_t \in b_t}\xi_{i_t}^\top\xi_{j_t} + \sum_{i_t \in b_t}\xi_{i_t}^\top\xi_{i_t}\right] - \frac{1}{n^2}\sum_{i,j=1}^{n}\xi_i^\top\xi_j$$

$$= \frac{1}{b^2}\left(\frac{b(b-1)}{n(n-1)}\sum_{i \neq j}\xi_i^\top\xi_j + \frac{b}{n}\sum_{i=1}^{n}\xi_i^\top\xi_i\right) - \frac{1}{n^2}\sum_{i,j=1}^{n}\xi_i^\top\xi_j$$

$$= \frac{1}{nb}\left(\frac{b-1}{n-1}\sum_{i,j=1}^{n}\xi_i^\top\xi_j + \left(1 - \frac{b-1}{n-1}\right)\sum_{i=1}^{n}\xi_i^\top\xi_i\right) - \frac{1}{n^2}\sum_{i,j=1}^{n}\xi_i^\top\xi_j$$

$$= \frac{n-b}{nb(n-1)}\left(-\frac{1}{n}\sum_{i,j=1}^{n}\xi_i^\top\xi_j + \sum_{i=1}^{n}\xi_i^\top\xi_i\right),$$

where the second equation is due to $\mathbb{E}\|\xi - \mathbb{E}\xi\|^2 = \mathbb{E}\|\xi\|^2 - \|\mathbb{E}\xi\|^2$.

Note that $\frac{1}{n}\sum_{i,j=1}^{n}\xi_i^\top\xi_j = n\|\frac{1}{n}\sum_{i=1}^{n}\xi_i\|^2 \geq 0$. Combing the above result with Lemma 6 we obtain that

$$\mathbb{E}_t\|\boldsymbol{d}_t - \nabla F(\boldsymbol{w}_t)\|^2 \leq \frac{n-b}{bn(n-1)}\sum_{i=1}^{n}\|\nabla f_i(\boldsymbol{w}_t) - \nabla f_i(\bar{\boldsymbol{w}})\|^2$$

$$\leq \frac{n-b}{bn(n-1)}\sum_{i=1}^{n}\left(\|\nabla f_i(\boldsymbol{w}_t) - \nabla f_i(\boldsymbol{w}_\star)\|^2 + \|\nabla f_i(\bar{\boldsymbol{w}}) - \nabla f_i(\boldsymbol{w}_\star)\|^2\right)$$

$$\leq \frac{2(n-b)L}{b(n-1)}\left(P(\boldsymbol{w}_t) - P(\boldsymbol{w}_\star) + P(\bar{\boldsymbol{w}}) - P(\boldsymbol{w}_\star)\right).$$

This completes the proof. $\qquad\square$

**Lemma 9** *Suppose Assumption 2 holds and $2\alpha L\frac{n-b}{b(n-1)} < 1$. Let $\boldsymbol{w}_\star = \arg\min_{\boldsymbol{w}} P(\boldsymbol{w})$. Then we obtain an output $\bar{\boldsymbol{w}}_s$ of Algorithm 3 such that*

$$\mathbb{E}P(\bar{\boldsymbol{w}}_s) - P(\boldsymbol{w}_\star) \leq \rho^s\left(P(\bar{\boldsymbol{w}}_0) - P(\boldsymbol{w}_\star)\right),$$

*where $\rho = \frac{U}{D}$, $U = 2\alpha m\left(1 - 2\alpha L\frac{n-b}{b(n-1)}\right)$ and $D = \left(\frac{2}{\mu_h} + 4\alpha^2 L(m+1)\frac{n-b}{b(n-1)}\right)$.*

*Proof:* We first consieder the $s$-th outer iteration of Algorithm 3. We have that $\bar{\boldsymbol{w}} = \bar{\boldsymbol{w}}_{s-1}$. We define $\widetilde{\boldsymbol{w}}_{t+1} = \text{Prox}_{\alpha h}\left(\boldsymbol{w}_t - \alpha\nabla F(\boldsymbol{w}_t)\right)$, which is independent of the mini-batch $b_t$. First we bound the last term in (24):

$$- 2\alpha\Delta_t^\top(\boldsymbol{w}_{t+1} - \boldsymbol{w}_\star) = -2\alpha\Delta_t^\top(\boldsymbol{w}_{t+1} - \widetilde{\boldsymbol{w}}_{t+1} + \widetilde{\boldsymbol{w}}_{t+1} - \boldsymbol{w}_\star)$$
$$\leq 2\alpha\|\Delta_t\|\|\boldsymbol{w}_{t+1} - \widetilde{\boldsymbol{w}}_{t+1}\| - 2\alpha\Delta_t^\top(\widetilde{\boldsymbol{w}}_{t+1} - \boldsymbol{w}_\star)$$
$$\leq 2\alpha\|\Delta_t\|\|\boldsymbol{w}_t - \alpha\boldsymbol{d}_t - \boldsymbol{w}_t + \alpha\nabla F(\boldsymbol{w}_t)\| - 2\alpha\Delta_t^\top(\widetilde{\boldsymbol{w}}_{t+1} - \boldsymbol{w}_\star)$$
$$= 2\alpha^2\|\Delta_t\|^2 - 2\alpha\Delta_t^\top(\widetilde{\boldsymbol{w}}_{t+1} - \boldsymbol{w}_\star), \tag{25}$$

where in the second inequality we use the non-expansiveness of the projection operator. Note that $\mathbb{E}_t\Delta_t^\top(\widetilde{\boldsymbol{w}}_{t+1} - \boldsymbol{w}_\star) = (\mathbb{E}_t\Delta_t)^\top(\widetilde{\boldsymbol{w}}_{t+1} - \boldsymbol{w}_\star) = 0$. Taking the conditional expectation $\mathbb{E}_t$ on both sides of (25) and using Lemma 8 we obtain that

$$-2\alpha\mathbb{E}_t\Delta_t^\top(\boldsymbol{w}_{t+1} - \boldsymbol{w}_\star) \leq \frac{4\alpha^2(n-b)L}{b(n-1)}\left(P(\boldsymbol{w}_t) - P(\boldsymbol{w}_\star) + P(\bar{\boldsymbol{w}}_{s-1}) - P(\boldsymbol{w}_\star)\right).$$

Taking the conditional expectation $\mathbb{E}_t$ on both sides of (24) and plugging in the above result, we obtain that

$$\mathbb{E}_t\|\boldsymbol{w}_{t+1} - \boldsymbol{w}_\star\|^2 \leq \|\boldsymbol{w}_t - \boldsymbol{w}_\star\|^2 - 2\alpha(\mathbb{E}_tP(\boldsymbol{w}_{t+1}) - P(\boldsymbol{w}_\star))$$
$$+ \frac{4\alpha^2(n-b)L}{b(n-1)}\left(P(\boldsymbol{w}_t) - P(\boldsymbol{w}_\star) + P(\bar{\boldsymbol{w}}_{s-1}) - P(\boldsymbol{w}_\star)\right). \tag{26}$$

Taking the expectation on both sides of (26), using the law of total expectation and summing over $t = 0, \ldots, m-1$ we obtain that

$$\mathbb{E}\|\boldsymbol{w}_m - \boldsymbol{w}_\star\|^2 + 2\alpha(\mathbb{E}P(\boldsymbol{w}_m) - P(\boldsymbol{w}_\star)) + 2\alpha\left(1 - 2\alpha L\frac{n-b}{b(n-1)}\right)\sum_{t=1}^{m-1}(\mathbb{E}P(\boldsymbol{w}_t) - P(\boldsymbol{w}_\star))$$

$$\leq \|\boldsymbol{w}_0 - \boldsymbol{w}_\star\|^2 + 4\alpha^2 L\frac{n-b}{b(n-1)}\left(P(\boldsymbol{w}_0) - P(\boldsymbol{w}_\star)\right) + 4\alpha^2 Lm\frac{n-b}{b(n-1)}\left(P(\bar{\boldsymbol{w}}_{s-1}) - P(\boldsymbol{w}_\star)\right). \tag{27}$$

Since $\boldsymbol{w}_0 = \bar{\boldsymbol{w}}_{s-1}$ and $2\alpha \geq 2\alpha\left(1 - 2\alpha L\frac{n-b}{b(n-1)}\right)$ by the assumption, we obtain that

$$\mathbb{E}\|\boldsymbol{w}_m - \boldsymbol{w}_\star\|^2 + 2\alpha\left(1 - 2\alpha L\frac{n-b}{b(n-1)}\right)\sum_{t=1}^{m}(\mathbb{E}P(\boldsymbol{w}_t) - P(\boldsymbol{w}_\star))$$

$$\leq \|\bar{\boldsymbol{w}}_{s-1} - \boldsymbol{w}_\star\|^2 + 4\alpha^2 L(m+1)\frac{n-b}{b(n-1)}\left(P(\bar{\boldsymbol{w}}_{s-1}) - P(\boldsymbol{w}_\star)\right). \tag{28}$$

By the $\mu_h$-strong convexity of $P$ and the definition of $\bar{w}_s$, we obtain that

$$2\alpha m \left( 1 - 2\alpha L \frac{n-b}{b(n-1)} \right) (\mathbb{E}P(\bar{w}_s) - P(w_\star))$$

$$\leq \left( \frac{2}{\mu_h} + 4\alpha^2 L(m+1) \frac{n-b}{b(n-1)} \right) (P(\bar{w}_{s-1}) - P(w_\star)).$$

Finally, by applying the above inequality recursively, we get the desired result. $\qquad\square$

**Corollary 3** *Let $w_\star = \arg\min_w P(w)$. With the same conditions as Lemma 9, setting the step size $\alpha = \frac{b(n-1)}{5L(n-b)}$ and the loop length $m = \frac{100L(n-b)}{\mu_h b(n-1)} + 8$, we obtain an output $w_s$ of Algorithm 3 such that $\mathbb{E}P(w_s) - P(w_\star) \leq \epsilon$ in a total sample complexity of $O\left( \left( n + \frac{(n-b)L}{(n-1)\mu} \right) \log \frac{1}{\epsilon} \right)$.*

*Proof:* Through simple calculations, we can obtain that $\rho = \frac{3}{4}$. Thus Algorithm 3 has geometric convergence. We need $s \geq \log \frac{4}{3} \log \frac{P(w_0) - P(w_\star)}{\epsilon}$ to obtain an $\epsilon$-optimal solution in expectation. The total sample complexity is $s(n + bm) = O\left( \left( n + \frac{(n-b)L}{(n-1)\mu} \right) \log \frac{1}{\epsilon} \right)$. $\qquad\square$

Now we are ready to prove the main result for SOREL with mini-batching based on Theorem 1.

**Proof of Corollary 2**

*Proof:* By Corollary 3, we get a $\delta_k$ approximate solution of the $k$-th outer loop of SOREL with mini-batching with the sample complexity of $O\left( \left( n + \frac{(n-b)L}{(k+1)(n-1)\mu} \right) \log \delta_k^{-1} \right)$. Similar to the proof of Corollary 1, through simple calculations we obtain the total sample complexity of $O\left( \frac{n^{\frac{3}{2}} G}{\mu \sqrt{\epsilon}} \log \frac{1}{\epsilon} + \frac{L(n-b)}{\mu(n-1)} \left( \log \frac{1}{\epsilon} \right)^2 \right).$

$\qquad\square$

## E  ADDITIONAL EXPERIMENTS ON NONCONVEX OBJECTIVES

In this section, we empirically explore the performance of SOREL in optimizing nonlinear models. We train a two-layer neural network with ReLU activation function and set the hidden layer's dimension equal to the feature dimension of the input data. We use the regression task from Section 5.1. Thus, the loss function and the model can be written as

$$\ell(z) = \frac{1}{2}(z - y)^2$$

and

$$z = W_2 \left( \text{ReLU}(W_1 x + b_1) \right) + b_2,$$

where $x \in \mathbb{R}^d, y \in \mathbb{R}$ are the feature and label, $W_1 \in \mathbb{R}^{d \times d}$, $b_1 \in \mathbb{R}^d$, $W_2 \in \mathbb{R}^{1 \times d}$ and $b_2 \in \mathbb{R}$ are trainable parameters. The experimental setup is identical to that in Section 5.1. The initial points are generated from a Gaussian distribution, and for the same spectral risk setting, the initial points for all algorithms are the same.

Figure 7 shows the training curves using three spectral risk measures on the `energy` and `concrete` datasets. Note that none of the four algorithms have theoretical guarantees in the non-convex setting. However, SOREL achieves the optimal or near-optimal results across various settings. On the `energy` dataset, SOREL achieves the lowest losses, significantly outperforming LSVRG and Prospect. On the `concrete` dataset, SOREL also achieves slightly lower loss values compared to LSVRG and Prospect. Table 4 reports the mean losses and standard deviations over the last ten passes. We observe that SOREL achieves the lowest mean losses, demonstrating the effectiveness in optimizing non-convex functions, even though theoretical guarantees are not available.

## F  FURTHER DISCUSSION ON THE COMPLEXITY IN COROLLARY 1

In Section 4, we discussed the optimality of the complexity in Corollary 1 with respect to $\epsilon$. In this section, we further discuss the dependence of the complexity in the Corollary 1 on $n$. The sample

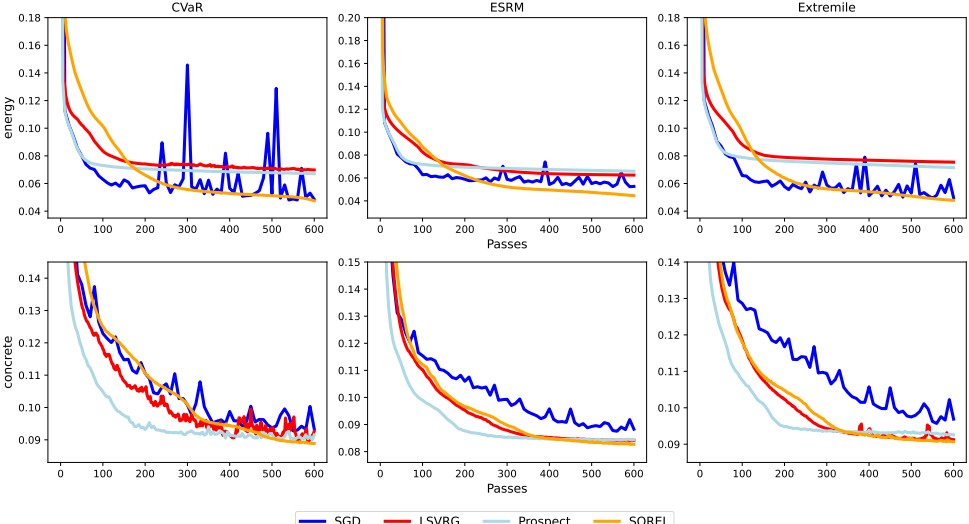

Figure 7: Results of training two-layer neural networks using different algorithms.

Table 4: The mean function values over the last ten passes (and standard deviations in parentheses).

| Datasets | energy | | | | concrete | | | |
|---|---|---|---|---|---|---|---|---|
| | SGD | LSVRG | Prospect | SOREL | SGD | LSVRG | Prospect | SOREL |
| CVaR | 0.05533 (0.02004) | 0.06986 (0.00027) | 0.06715 (0.00024) | 0.04787 (0.00045) | 0.09573 (0.00299) | 0.09149 (0.00276) | 0.09099 (0.00188) | 0.08888 (0.00104) |
| ESRM | 0.05463 (0.01268) | 0.06253 (0.00882) | 0.06594 (0.00006) | 0.04464 (0.00020) | 0.08942 (0.00253) | 0.08417 (0.00006) | 0.08443 (0.00039) | 0.08275 (0.00013) |
| Extremile | 0.05153 (0.00329) | 0.07544 (0.00005) | 0.07155 (0.00005) | 0.04792 (0.00005) | 0.09832 (0.00325) | 0.09132 (0.00012) | 0.09261 (0.00097) | 0.09073 (0.00077) |

complexity with respect to $n$ and $\epsilon$ in Corollary 1 is $\widetilde{O}(n^{3/2}/\sqrt{\epsilon})$. Since SOREL requires computing the projection onto the permutahedron $\Pi_{\boldsymbol{\sigma}}$ in each outer iteration, which takes $O(n \log n)$ time, the total time complexity of SOREL includes an additional $O(Kn \log n)$ term, where $K = O\left(\frac{\sqrt{n}G}{\mu\sqrt{\epsilon}}\right)$ is given in the proof of Corollary 1. Therefore, the total complexity of SOREL with respect to $n$ and $\epsilon$ is $\widetilde{O}(n^{3/2}/\sqrt{\epsilon})$.

We then discuss the complexity of baselines in Section 5 with respect to $n$ and $\epsilon$. SGD has been shown to fail to converge to the optimal solution of the spectral risk minimization problem, while LSVRG only guarantees convergence for $\nu \geq \Omega\left(nG^2/\mu\right)$ (Mehta et al., 2022). For Prospect, by setting $\nu = O(\epsilon)$, we obtain its sample complexity with respect to $n$ and $\epsilon$ as $\widetilde{O}(n^2/\epsilon^2)$ or $\widetilde{O}(n/\epsilon^3)$ (depending on the size of $\epsilon$). Moreover, Prospect requires computing the projection onto the permutahedron $\Pi_{\boldsymbol{\sigma}}$ at each step with a cost of $O(n \log n)$ time, which results in its total time complexity an additional $\widetilde{O}(n^2/\epsilon^3)$ or $\widetilde{O}(n^3/\epsilon^2)$. Therefore, SOREL also has an advantage in terms of the total time complexity with respect to $n$. This is consistent with the experimental results in Section 5, where SOREL significantly outperforms Prospect in terms of runtime.

Beyond the above, after our original submission of this work we have also recently become aware that Mehta et al. (2024a) proposed a primal-dual stochastic algorithm to solve the smoothed version of the spectral risk minimization problem. By setting the smoothing parameter $\nu$ to $\epsilon$, its sample complexity is $O\left(\frac{n}{b}\sqrt{\frac{nG^2}{\mu\epsilon}}\ln\left(\frac{1}{\epsilon}\right)\right)$, where $b$ is the batchsize. Considering that each step of their algorithm requires a projection, the total time complexity includes an additional $\widetilde{O}\left(\frac{n^{2.5}}{b\sqrt{\epsilon}}\right)$. In the latest manuscript of Mehta et al. (2024a), their improved algorithm can also solve the original spectral risk minimization problem. Their theoretical analysis focuses on deriving a linear convergence rate for the smoothed specral risk minimization problem and does not provide a convergence rate for the original problem.

