# OpenReview forum: "SOREL: A Stochastic Algorithm for Spectral Risks Minimization"
_ICLR.cc/2025/Conference — ICLR 2025 Poster_

### Official Review · Reviewer_TvDg · 2024-10-31

**Soundness:** 3
**Presentation:** 3
**Contribution:** 3
**Rating:** 6
**Confidence:** 2

**Summary:**

This paper introduces SOREL, a stochastic gradient-based algorithm for spectral risk minimisation. After an introduction and an overview of this work, the authors carefully present the current drawbacks of current approaches of spectral risk minimisation alongside their new algorithm in Section 3. Section 4 provide convergence guarantees, and Section 5 gathers various experiments showing competitive performance and computational time with respect to existing methods.

**Strengths:**

- Sections 1, 2 and 3.1 are truly well-written, it has been a pleasure to read it
- The idea of using a primal dual method based on Lemma 1 of (Blondel et al. 2020) is clever and elegant
- Algorithmic performances are convincing and are accompanied by sound theory
- It is great to propose a mini batching version of SOREL

**Weaknesses:**

See Questions part

**Questions:**

- You wrote in your abstract that there is a lack of guarantees for the original spectral risk. You said later in l.78 that SOREL is the first stochastic algorithm with convergence guarantees for the spectral risk minimisation problem. However, in Section 3.1 you directly consider a regularised version of the spectral risk with a strongly convex function (Eq. 1). Later, in Eq. (4) you added another regulariser in $\mathbf{\lambda}$ to avoid huge variations of $\lambda$ during the learning process. All of these reason suggests that you actually do NOT deal with the original spectral risk. Do I miss something?
- l. 268-269 : Those lines are unclear, I do not under whether there is a need to compute $\mathcal{O}(n\log(n))$ operations at line 5, within the loop of line 3 in Algorithm 1 or if this is the total number of computations. Can you explicitly write the time complexity of your algorithm and compare it with those proposed in literature? This would draw a useful theoretical link to understand why your algorithm is faster than others in Section 5.
- I am not sure to understand why Lemma 1 is stated here, as there is no proof sketch in the main document. What do I miss?
- In Theorem 1, can you please define $w^*$? I do not understand what it is.
- In Section 5, it seems that you focused exclusively on linear regression, would SOREL work with small neural nets with 1-2 hidden layers? Even if theoretical guarantees may not hold anymore, I am curious to know whether the methods remain tractable and efficient.

---

> ### Author Response · Authors · 2024-11-19
>
> We thank reviewer TvDg for the constructive feedback.
>
> ## 1. Not deal with the original spectral risk
>
> In Eq. (1), $g(\boldsymbol{w})$ represents the regularizer on the model parameters. In Eq. (2), we present the equivalent form of the original spectral risk as $R_{\boldsymbol{\sigma}}(\boldsymbol{w})=\max_{\boldsymbol{\lambda} \in \Pi_{\boldsymbol{\sigma}}} \sum_{i=1}^n \lambda_i \ell_i(\boldsymbol{w})$. In lines 340-342 and 127-130, we point out that other methods smooth the original spectral risk by adding a strongly concave term with respect to $\boldsymbol{\lambda}$. Specifically, the smoothed spectral risk can be expressed as: $\tilde{R}\_{\boldsymbol{\sigma}}(\boldsymbol{w})=\max_{\boldsymbol{\lambda} \in \Pi_{\boldsymbol{\sigma}}} \sum_{i=1}^{n} \lambda_i \ell_i(\boldsymbol{w})-\nu\Omega(\boldsymbol{\lambda})$, where $-\nu\Omega(\boldsymbol{\lambda})$ is a strongly concave function with respect to $\boldsymbol{\lambda}$, and $\nu>0$. $\tilde{R}_{\boldsymbol{\sigma}}(\boldsymbol{w})$ is differentiable with respect to $\boldsymbol{w}$ [1], and by setting $\nu=O(\epsilon)$, the gap between the original spectral risk and the smoothed spectral risk does not exceed $\epsilon$.
>
> Therefore, in Eq. (1), we consider the original spectral risk. Eq. (1) can be viewed as minimizing the spectral risk with a regularization term on the model parameters, such as the L2 regularization. If we set $\sigma_i=1/n$, Eq. (1) simplifies to minimizing the empirical risk plus a regularization term on the model parameters.
>
> In Section 3.1, we explain that our algorithm can be viewed as iteratively solving Eq. (4) and Eq. (5). Therefore, Eq. (4) is merely a subproblem that needs to be solved during the execution of Algorithm 1. Algorithm 1 is still aimed at solving Eq. (1).
>
> We apologize for the confusion that may arise between the regularizer on the model parameter and the strong concave term with respect to $\boldsymbol{\lambda}$. We will include the formulation of the smoothed spectral risk in our paper to eliminate ambiguity.
>
> ##  2. Total time complexity
>
> As mentioned in Section 3.1, our algorithm can be seen as alternately solving Eq. (4) and Eq. (5). Solving Eq. (4) involves computing the projection onto the permutahedron $\Pi_{\boldsymbol{\sigma}}$, which can be done in $O(n\log n)$ time. Line 5 of Algorithm 1 is inside the outer loop in Line 3, so the total time complexity requires an additional $O(Kn\log n)$, where $K$ is given in Corollary 1 as $K=O\left(\frac{\sqrt{n} G}{\mu \sqrt{\epsilon}}\right)$. Therefore, the total complexity of our algorithm with respect to $n$ and $\epsilon$ is $\tilde{O}(n^{3/2}/\sqrt{\epsilon})$.
>
> Next, we compare this complexity with baselines in Section 5. SGD has been shown to fail to converge to the optimal solution of the spectral risk minimization problem, and LSVRG only guarantees convergence for $\nu \geq \Omega\left(n G^2 / \mu\right)$ [2]. For Prospect, for any $\epsilon > 0$, the sample complexity to achieve an $O(\epsilon)$ optimal solution is $\tilde{O}(n/\epsilon^3)$ or $\tilde{O}(n^2/\epsilon^2)$ (depending on the size of $\epsilon$). However, Prospect requires solving a projection problem similar to Eq. (4) at each step, so the total time complexity is $\tilde{O}(n^2/\epsilon^3)$ or $\tilde{O}(n^3/\epsilon^2)$.
>
> We thank the reviewer for highlighting the importance of the total time complexity and the complexity with respect to $n$. We will include this discussion in our paper.
>
> ## 3. Understanding Lemma 1
>
> We apologize for the confusion caused here. A key idea of our proof is that $L\left(\boldsymbol{w}, \boldsymbol{\lambda}^{\star}\right) \geq L\left(\boldsymbol{w}^{\star}, \boldsymbol{\lambda}\right)$ for any $\boldsymbol{w} \in \mathbb{R}^d$ and $\boldsymbol{\lambda} \in \Pi_{\boldsymbol{\sigma}}$, where $\boldsymbol{w}^\star$ and $\boldsymbol{\lambda}^\star$ are an optimal solution to Eq. (3). Indeed, we know that $L\left(\boldsymbol{w}^{\star}, \boldsymbol{\lambda}^{\star}\right)=\max_{\boldsymbol{\lambda} \in \Pi_\sigma} L\left(\boldsymbol{w}^{\star}, \boldsymbol{\lambda}\right) \geq L\left(\boldsymbol{w}^{\star}, \boldsymbol{\lambda}\right)$ and $L\left(\boldsymbol{w}, \boldsymbol{\lambda}^{\star}\right) \geq L\left(\boldsymbol{w}^{\star}, \boldsymbol{\lambda}^{\star}\right)=\min _{\boldsymbol{w}} L\left(\boldsymbol{w}, \boldsymbol{\lambda}^{\star}\right)$. Thus, as mentioned before Theorem 1 in Section 4, we can choose appropriate parameters of Algorithm 1 such that the adjacent terms indexed by $k$ on the right-hand side of Eq. (6) can be canceled out during summation, and the left-hand side of Eq. (6) is always greater than or equal to 0.
>
> Due to space limits, we did not include this idea in the main text. We will adjust the content of Section 4 in future revisions to make the theoretical analysis more readable.

---

> ### Author Response · Authors · 2024-11-19
>
> ## 4. Definition of $\boldsymbol{w}^\star$
>
> We apologize for not formally defining $\boldsymbol{w}^\star$. $\boldsymbol{w}^\star$ should be defined as the optimal solution to Eq. (1). Since the objective function in Eq. (1) is strongly convex under Assumption 1, $\boldsymbol{w}^\star$ is unique. We have added this definition to Theorem 1 in the revised manuscript.
>
> ## 5. Training small neural nets
>
> We train a two-layer neural network with ReLU activation function. We conduct experiments on the same regression task as in Section 5.1, on the energy and concrete real-world datasets. Detailed experimental results are presented in Appendix F of our revised manuscript. Here, we report the mean losses and standard deviations (in parentheses) over the last ten passes. We observe that our method outperforms other baselines, especially on the energy dataset, where our method achieves significantly lower losses, even though none of the four algorithms have theoretical guarantees.
>
> **Energy**
>
> |           | SGD                | LSVRG             | Prospect           | SOREL              |
> | --------- | ------------------ | ----------------- | ------------------ | ------------------ |
> | CVaR      | 0.05533  (0.02004) | 0.06986 (0.00027) | 0.06715 (0.00024)  | 0.04787 (0.00045)  |
> | ESRM      | 0.05463 (0.01268)  | 0.06253 (0.00882) | 0.06594 (0.00006)  | 0.04464  (0.00020) |
> | Extremile | 0.05153 (0.00329)  | 0.07544 (0.00005) | 0.07155  (0.00005) | 0.04792 (0.00005)  |
>
> **Concrete**
>
> |           | SGD               | LSVRG              | Prospect            | SOREL              |
> | --------- | ----------------- | ------------------ | ------------------- | ------------------ |
> | CVaR      | 0.09573 (0.00299) | 0.09149 (0.00276)  | 0.09099  (0.00188)  | 0.08888  (0.00104) |
> | ESRM      | 0.08942 (0.00253) | 0.08417  (0.00006) | 0.08443 (0.00039)   | 0.08275 (0.00013)  |
> | Extremile | 0.09832 (0.00325) | 0.09132 (0.00012)  | 0.09261   (0.00097) | 0.09073 (0.00077)  |
>
>
>
> ## References
>
> [1] Nesterov, Yu. "Smooth minimization of non-smooth functions." *Mathematical programming* 103 (2005): 127-152.
>
> [2] Mehta, Ronak, et al. "Stochastic optimization for spectral risk measures." *International Conference on Artificial Intelligence and Statistics*. PMLR, 2023.

---

> > ### Comment · Reviewer_TvDg · 2024-11-25
> > **Thank you**
> >
> > Thank you for your answer, which address my concerns. I am maintaining my current score.

---

### Official Review · Reviewer_YNq6 · 2024-11-03

**Soundness:** 1
**Presentation:** 4
**Contribution:** 2
**Rating:** 6
**Confidence:** 5

**Summary:**

The paper presents SOREL, an optimization algorithm designed for spectral risk measure objectives, which can also be understood as  primal-dual min-max objectives for learning problems. In comparison to previous stochastic methods, the algorithm adds a proximal term to the dual update and claims a near-optimal convergence guarantee for the strongly convex-(nonstrongly) concave variant of this objective. The method employs an SVRG-style variance reduction similar to [Mehta et al. (2023)](https://proceedings.mlr.press/v206/mehta23b.html). While SOREL generally performs favorably in experiments, the theoretical convergence analysis is incorrect, making this work unpublishable in its current state.

**Strengths:**

- The paper points out an important fact about the analysis of algorithms for "smoothed" spectral risk measures, which is that when the smoothing parameter $\nu = O(\epsilon)$, it should be included in the complexity guarantee, making "linearly convergent" methods sublinear for the original non-smooth problem.
- By using the experimental benchmark of [Mehta et al. (2024)](https://arxiv.org/pdf/2310.13863), the authors are able to perform head-to-head comparisons to  recent algorithms designed for spectral risk measures (Figure 3).  The empirical performance is competitive with, and often outperforms, other methods.
- The exposition is clear and well-written, although some sections are nearly identical to sections in the referenced work on spectral risk measure optimization with stochastic algorithms (Section 1, Section 2, Appx B).

**Weaknesses:**

Upon viewing Corollary 1, we see that the complexity contains the term $\log(\frac{\sqrt{n}G}{\mu^2\sqrt{\epsilon}})$, i.e. a logarithm is taken for a quantity that is *not* unitless. This does not pass a basic sanity check for convergence analyses in optimization theory: that the complexity does not depend on the units of the input. The error comes from the fact that the precision parameter $\delta_k$ in the inner loop is dependent on the strong convexity parameter $\mu$ (units of loss/inputs$^2$) when it should be in units of loss (based on Lemma 3). To see why this is invalid, recall that an optimization algorithm should not change its exact complexity when the inputs are reparametrized in different units. Consider the objective as a function of $\boldsymbol{w}/10$ instead of $\boldsymbol{w}$. Then, we have the Lipschitz and strong convexity constants change as $G \mapsto 10G$ and $\mu \mapsto 100\mu$. Then,
$$
\frac{\sqrt{n}(10G)}{(100\mu)^2\sqrt{\epsilon}} = \frac{1}{1000}\frac{\sqrt{n}G}{\mu^2\sqrt{\epsilon}}.
$$
This indicates that the number of iterations in the inner loop would change based on the scale, which is not a valid analysis. Please check the units before presenting the result.

Matters such as significance, novelty, and impact come second to the analysis, especially as the experimental results do not show a large practical improvement over the methods of [Mehta et al. (2023)](https://proceedings.mlr.press/v206/mehta23b.html) and [Mehta et al. (2024)](https://arxiv.org/pdf/2310.13863). Please correct the analysis.

**Questions:**

n/a

---

> ### Author Response · Authors · 2024-11-19
>
> We thank reviewer YNq6 for the thoughtful feedback, especially for pointing out that the complexity in Corollary 1 includes units.
>
> ## 1. Complexity not unitless
>
> We acknowledge that the complexity in Corollary 1 is not unitless.
>
> We point out that the unit in the complexity arises from the use of the inequality $2\left(\mathbb{E}\|\boldsymbol{x}-\overline{\boldsymbol{x}}\|^2\right)^{\frac{1}{2}} \leq \left(1+\mathbb{E}\|\boldsymbol{x}-\overline{\boldsymbol{x}}\|^2\right)$ in the proof of Lemma 2 at line 803. While this inequality is mathematically correct, it results in inconsistent units on both sides of the inequality. To ensure consistent units, we simply need to modify the inequality to $2\left(\mathbb{E}\|\boldsymbol{x}-\overline{\boldsymbol{x}}\|^2\right)^{\frac{1}{2}} \leq \left(D+D^{-1}\mathbb{E}\|\boldsymbol{x}-\overline{\boldsymbol{x}}\|^2\right)$, where $D=G/\mu$. Next, we only need to adjust the coefficients of the affected terms in our analysis and set $\tilde{\delta}_k=D^2\delta_k$ in Theorem 1, where $\delta_k=\min \left(\frac{\mu}{8(k+5)}, \mu(k+1)^{-6}\right)$  is the parameter used in Theorem 1 when we obtained the complexity with units. This allows us to obtain the unitless complexity $O\left(\frac{n^{\frac{3}{2}} G}{\mu \sqrt{\epsilon}} \log \frac{\sqrt{n}}{G \sqrt{\epsilon}}+\frac{L}{\mu} \log \frac{\sqrt{n}}{G \sqrt{\epsilon}} \log \frac{\sqrt{n} G}{\mu \sqrt{\epsilon}}\right)$. Note that the numerator in $\log \frac{\sqrt{n}}{G \sqrt{\epsilon}}$ implicitly includes units of loss. Indeed, the modified precision parameter $\tilde{\delta}_k$ in the inner loop is in units of loss.
>
> We have uploaded a revised version of the manuscript and corrected our analysis, including Theorem 1 and Corollary 1 in the main text, as well as the proof in Appendix 1. The changes are highlighted in blue. We emphasize that our proof before modification is mathematically correct. To obtain a unitless complexity, we only need to adjust the coefficients of certain terms, and our analysis remains valid.
>
> ## 2. No large practical improvements in experimental results
>
> We state that our main contribution is the proposal of the first stochastic algorithm for solving the original spectral risk minimization problem, achieving a near-optimal complexity with respect to $\epsilon$. The convergence of SGD and LSVRG lacks theoretical guarantees. In our experiments, we primarily focus on the performance of the algorithm during the training process, rather than pursuing state-of-the-art test metrics. In the experiments, SOREL successfully converges to the true optimal solution under various spectral risk settings, with optimal or near-optimal runtime. This aligns with the discussions with reviewer TvDg and reviewer fKff, where we point out that SOREL also has advantages in complexity with respect to the sample size $n$.
>
> Additionally, following the suggestions of reviewer fKff and reviewer TvDg, we train a two-layer neural network under the experimental setup of Section 5.1. Detailed experimental results are presented in Appendix F of our revised manuscript. Here, we report the mean losses and standard deviations (in parentheses) over the last ten passes. We observe that our method outperforms other baselines, especially on the energy dataset, where our method achieves significantly lower losses, even though none of the four algorithms have theoretical guarantees.
>
> **Energy**
>
> |           | SGD                | LSVRG             | Prospect           | SOREL              |
> | --------- | ------------------ | ----------------- | ------------------ | ------------------ |
> | CVaR      | 0.05533  (0.02004) | 0.06986 (0.00027) | 0.06715 (0.00024)  | 0.04787 (0.00045)  |
> | ESRM      | 0.05463 (0.01268)  | 0.06253 (0.00882) | 0.06594 (0.00006)  | 0.04464  (0.00020) |
> | Extremile | 0.05153 (0.00329)  | 0.07544 (0.00005) | 0.07155  (0.00005) | 0.04792 (0.00005)  |
>
> **Concrete**
>
> |           | SGD               | LSVRG              | Prospect            | SOREL              |
> | --------- | ----------------- | ------------------ | ------------------- | ------------------ |
> | CVaR      | 0.09573 (0.00299) | 0.09149 (0.00276)  | 0.09099  (0.00188)  | 0.08888  (0.00104) |
> | ESRM      | 0.08942 (0.00253) | 0.08417  (0.00006) | 0.08443 (0.00039)   | 0.08275 (0.00013)  |
> | Extremile | 0.09832 (0.00325) | 0.09132 (0.00012)  | 0.09261   (0.00097) | 0.09073 (0.00077)  |

---

> ### Author Response · Authors · 2024-11-19
>
> ## 3. Similarity to referenced work
>
> We appreciate the reviewer's recognition of our paper's presentation in the Strengths section, and we thank the reviewer for pointing out the similarity in writing between our work and some referenced work.
>
> Here we would like to clarify some portions that may be similar to referenced work. Since spectral risk measures are not yet widely known in the machine learning field, introducing them from the perspective of the empirical risk is natural and intuitive. This perspective of introduction is similar across works related to the spectral risk research. Additionally, we have elaborated on our research motivation in detail in Section 1.
>
> In Section 2, we review the existing work from the perspective of the spectral risk measure, particularly focusing on deterministic and stochastic methods for solving the spectral risk minimization problem. We also specifically highlight and analyze the shortcomings of existing stochastic algorithms: their lack of convergence guarantees for the original spectral risk minimization problem. Our introduction to the applications of spectral risks may be similar to some references. But we interpret these applications from a unified perspective of spectral risks.
>
> We elaborate on the experimental details in Appendix B. Since we follow the same experimental setup as [1] in Section 5.1, we describe the experimental setup of [1] in Appendix B.  We have revised Appendix B to reduce redundancy with the experimental setup in [1].
>
> ## References
>
> [1] Mehta, Ronak, et al. "Distributionally robust optimization with bias and variance reduction."  The Twelfth International Conference on Learning Representations, abs/2310.13863, 2024.

---

> ### Comment · Reviewer_YNq6 · 2024-11-26
> **Response to Rebuttal**
>
> Thank you for your hard work and revision in the rebuttal period. I have raised my score to recommend acceptance.

---

### Official Review · Reviewer_fKff · 2024-11-03

**Soundness:** 3
**Presentation:** 3
**Contribution:** 3
**Rating:** 6
**Confidence:** 3

**Summary:**

This paper considers the problem of optimization in *spectral risk minimization*, which aims to produce more risk-sensitive models than standard ERM.  More precisely, spectral risk is defined as a weighted empirical risk minimization problem, where the weights depend on the order statistics of a given empirical sample, thereby subsuming both empirical risk (with all weights being the same) and such risk-sensitive measures as CVaR and Extremiles.  While minimizing spectral risk can be attractive due to the risk-sensitivity, it can be challenging to scale as the evaluation of the spectral risk of a given parameter depends on the entire dataset through the order statistics and is thus not amenable to naive applications of existing SGD approaches.  In particular, in order to obtain an unbiased estimate of the subgradient of the loss, all samples must have losses evaluated.  The authors circumvent this problem by applying the rearrangement inequality to reduce the problem to finding the value of a min-max game and then propose applying what amounts to a regularized form of projected gradient ascent-descent with momentum.  The authors present rates on their algorithm that are close to earlier lower bounds and conclude the paper with a number of experiments with ridge regression on a variety of instantiations of spectral risks and datasets, demonstrating that their approach is generally superior to alternatives.

**Strengths:**

This paper sets out an interesting problem that has some (at least distant) applications to relevant areas in ML including risk sensitivity, fairness, and generalization OOD.  The proposed solution of reducing to a minimax game and then running a primal-dual algorithm is interesting and the empirical validation uses a reasonable diversity of datasets and loss functions that suffices to convince me that the proposed algorithm is superior to alternatives, at least when considering ridge regression.

**Weaknesses:**

There are several main weaknesses with the paper:

1. With respect to the theory, I think more discussion of the strongly convex regularizer is necessary.  I understand that it is necessary in order to ensure identifiability, but I find the discussion under Corollary 1 confusing.  For example, in the comparison to related work, the authors cite several works that set the regularization term to be $O(\epsilon)$ in order to be $\epsilon$-suboptimal with respect to the unregularized solution and note that this leads to a worse sample complexity of $O(1/\epsilon)$; I am confused as to why the current proposal does not suffer the same drawback.  Indeed, in order to be $\epsilon$-suboptimal with respect to the unperturbed spectral risk, it seeems as if the bound in Corollary 1 would scale with $\epsilon^{-3/2}$ which is obviously worse than the alternative.

2. I think a more thorough discussion of the computational complexity of SOREL is in order.  While I understand that for most steps, the gradient updates are cheap and independent of $n$, lines 2, 7, and 11 of Algorithm 1 require linear in the number of samples time.  The authors claim that these steps are for variance reduction, but do not investigate the necessity thereof.  Another way of saying this is that the authors claim almost optimality of their algorithm in how the iteration complexity scales with $\epsilon$ but it seems like the dependence on $n$ is also fairly important (and is what motivates the study itself) and the extent to which the authors' algorithm is optimal in this respect is left largely undiscussed.

3. The experiments are all restricted to linear models.  While I understand that the authors wish to ensure that their key Assumption 1 is satisfied, it would be nice if they considered something nonlinear as well.  This is especially true because they remove the regularization in their experiments anyhow and thus are at least somewhat happy to depart from their theory.

**Questions:**

1. Can you define `total sample complexity' in Corollary 1 please?  Is this the sum of calls to a stochastic gradient oracle?  It would be helpful if this were made rigorous as my understanding of the key contribution of the present work is that the dependence in complexity on the number of samples is reduced substantially.

2. See weaknesses 1 and 2.

---

> ### Author Response · Authors · 2024-11-19
>
> We are grateful for the diligent efforts of reviewer fKff in evaluating our paper and for the constructive feedback provided.
>
> ## 1. Discussion of the strongly convex regularizer
>
> In Lines 340-342 and 127-130, we claim that other methods smooth the original spectral risk by adding a strongly concave term with a coefficient $\nu$ with respect to $\boldsymbol{\lambda}$. Specifically, the original spectral risk can be expressed as Eq. (2): $R_{\boldsymbol{\sigma}}(\boldsymbol{w})=\max_{\boldsymbol{\lambda} \in \Pi_{\boldsymbol{\sigma}}} \sum_{i=1}^n \lambda_i \ell_i(\boldsymbol{w})$.
>
> The smoothed spectral risk can be expressed as: $\tilde{R}\_{\boldsymbol{\sigma}}(\boldsymbol{w})=\max_{\boldsymbol{\lambda} \in \Pi_{\boldsymbol{\sigma}}} \sum_{i=1}^{n} \lambda_i \ell_i(\boldsymbol{w})-\nu\Omega(\boldsymbol{\lambda})$ , where $-\nu\Omega(\boldsymbol{\lambda})$ is a strongly concave function with respect to $\boldsymbol{\lambda}$ and $\nu>0$.  $\tilde{R}_{\boldsymbol{\sigma}}(\boldsymbol{w})$ is differentiable with respect to $\boldsymbol{w}$ [1], and by setting $\nu=O(\epsilon)$, the gap between the original spectral risk and the smoothed spectral risk does not exceed $\epsilon$. However, our method does not smooth the original spectral risk, and therefore, $\nu$ does not appear in our iteration complexity.
>
> $\mu$ is the strong convexity coefficient of the regularizer $g$ on the model parameters. In this paper, we consider minimizing the spectral risk with a regularizer (Eq. (1)), similar to minimizing the empirical risk with a regularizer.
>
> We apologize for the confusion that may arise between the regularizer on the model parameter and the strong concave term with respect to $\boldsymbol{\lambda}$. We will include the formulation of the smoothed spectral risk in our paper to eliminate ambiguity.
>
> ## 2. Discussion of the computational complexity of SOREL
>
> In Eq.(3), we adopt a minimax reformulation and solve Eq. (1) by alternately solving Eq. (4) and Eq. (5). In line 2 of Algorithm 1, we update $\boldsymbol{\lambda}$, which requires $O(n\log n)$ time. In lines 7 and 11 of Algorithm 1, we use variance reduction techniques to accelerate the solution of Eq. (5). As a result, the subproblem Eq. (5) converges linearly. However, this makes the complexity linearly dependent on $n$. In other words, we trade off complexity with respect to $\epsilon$ for complexity with respect to $n$. The $\sqrt{n}$ term originates from Assumption 1, where we assume $\ell_i$ is $G$ Lipschitz continuous. Therefore, $\boldsymbol{\ell}$ is $\sqrt{n}G$ Lipschitz continuous.
>
> Here, we discuss the complexity of baselines in Section 5 with respect to $n$ and $\epsilon$. SGD has been shown to fail to converge to the optimal solution of the spectral risk minimization problem, while LSVRG only guarantees convergence for $\nu \geq \Omega\left(n G^2 / \mu\right)$ [2]. For Prospect, by setting $\nu = O(\epsilon)$, we obtain its sample complexity with respect to $n$ and $\epsilon$ as $\tilde{O}(n^2/\epsilon^2)$ or $\tilde{O}(n/\epsilon^3)$  (depending on the size of $\epsilon$). This is similar to our case, where there is a tradeoff between complexity in $n$ and $\epsilon$.
>
> Moreover, as mentioned in Section 5.1 and in the response to reviewer TvDg, Prospect requires solving a projection problem similar to Eq. (4) at each step, which takes $O(n\log n)$ time. Therefore, its total time complexity is $\tilde{O}(n^2/\epsilon^3)$ or $\tilde{O}(n^3/\epsilon^2)$.  Note that SOREL solves Eq. (4) only once in each outer loop. Therefore, the total time complexity of SOREL requires an additional $O(Kn\log n)$, where $K=O\left(\frac{\sqrt{n} G}{\mu \sqrt{\epsilon}}\right)$ as stated in Corollary 1. Thus, the total time complexity of SOREL is $\tilde{O}(n^{3/2}/\sqrt{\epsilon})$.
>
> We thank the reviewer for pointing out the importance of $n$ in the complexity analysis, and we will include this discussion in our paper.

---

> ### Author Response · Authors · 2024-11-19
>
> ## 3. Experiments restricted to linear models
>
> As mentioned earlier, our method does not require adding a regularization term with respect to $\boldsymbol{\lambda}$ to the objective function.  We dropped the strongly concave term $-\nu\Omega(\boldsymbol{\lambda})$ required by the LSVRG and Prospect, as they have been observed to exhibit linear convergence for the original spectral risk minimization problem in practice even without it [3]. Please let us know if we have misunderstood anything.
>
> For nonlinear models, we train a two-layer neural network with ReLU activation function. We perform experiments on the same regression task as in Section 5.1, on the energy and concrete real-world datasets. Detailed experimental results are presented in Appendix F of our revised manuscript. Here, we present the mean losses and standard deviations (in parentheses) over the last ten passes of the training process. We find that SOREL outperforms other baselines, particularly on the energy dataset, where our method achieves significantly lower losses, even though none of the four algorithms have theoretical guarantees.
>
> **Energy**
>
> |           | SGD                | LSVRG             | Prospect           | SOREL              |
> | --------- | ------------------ | ----------------- | ------------------ | ------------------ |
> | CVaR      | 0.05533  (0.02004) | 0.06986 (0.00027) | 0.06715 (0.00024)  | 0.04787 (0.00045)  |
> | ESRM      | 0.05463 (0.01268)  | 0.06253 (0.00882) | 0.06594 (0.00006)  | 0.04464  (0.00020) |
> | Extremile | 0.05153 (0.00329)  | 0.07544 (0.00005) | 0.07155  (0.00005) | 0.04792 (0.00005)  |
>
> **Concrete**
>
> |           | SGD               | LSVRG              | Prospect            | SOREL              |
> | --------- | ----------------- | ------------------ | ------------------- | ------------------ |
> | CVaR      | 0.09573 (0.00299) | 0.09149 (0.00276)  | 0.09099  (0.00188)  | 0.08888  (0.00104) |
> | ESRM      | 0.08942 (0.00253) | 0.08417  (0.00006) | 0.08443 (0.00039)   | 0.08275 (0.00013)  |
> | Extremile | 0.09832 (0.00325) | 0.09132 (0.00012)  | 0.09261   (0.00097) | 0.09073 (0.00077)  |
>
> ## 4. Definition of `total sample complexity'
>
> Total sample complexity refers to the total number of samples used in Algorithm 1. The complexity of the gradient evaluation is the same as the complexity presented in Corollary 1. Indeed, in lines 8-15 of Algorithm 1, for every $m_k$ updates of $\boldsymbol{w}$, $(n + m_k)$ samples are required, as well as $(n + m_k)$ gradient evaluations (if $\nabla\ell_1(\bar{\boldsymbol{w}}), \ldots, \nabla\ell_n(\bar{\boldsymbol{w}})$ are stored) or $(n + 2m_k)$ gradient evaluations (if $\nabla\ell_1(\bar{\boldsymbol{w}}), \ldots, \nabla\ell_n(\bar{\boldsymbol{w}})$ are not stored).
>
> ## **References**
>
> [1] Nesterov, Yu. "Smooth minimization of non-smooth functions." *Mathematical programming* 103 (2005): 127-152.
>
> [2] Mehta, Ronak, et al. "Stochastic optimization for spectral risk measures." *International Conference on Artificial Intelligence and Statistics*. PMLR, 2023.
>
> [3] Mehta, Ronak, et al. "Distributionally robust optimization with bias and variance reduction."  The Twelfth International Conference on Learning Representations, abs/2310.13863, 2024.

---

> > ### Comment · Reviewer_fKff · 2024-11-19
> > **Response to Rebuttal**
> >
> > I appreciate the authors taking the time to respond to my questions.  While I appreciate the focus on the dependence on $\epsilon$, I think it would be good to include some mention of the dependence on $n$ in the paragraph starting at line 337 in the new draft.  I appreciate the clarification on the strongly convex regularizer as well and in light of this, but I am a little confused by the results of the neural network experiments.  Why does SGD outperform Prospect on all of the metrics on the Energy dataset, when prospect is specifically designed to minimize spectral risk?  I understand that this is not the primary focus of the paper and there are not rigorous guarantees for either of these algorithms in this setting, but is it not concerning that the primary empirical benchmark (Prospect) is not outperforming the method that is claimed to not work very well (SGD) for this problem?

---

> > > ### Author Response · Authors · 2024-11-20
> > > **Response to Reviewer fKff**
> > >
> > > We appreciate the reviewer’s timely response. It is surprising that Prospect performs worse than SGD on the energy dataset. We attempt to explain this phenomenon below.
> > >
> > > We first introduce the principle of SGD. As described in Section 1, the spectral risk assigns weights $\sigma_i$ to the loss of each sample: $R_{\boldsymbol{\sigma}}(\boldsymbol{w})=\sum_{i=1}^n \sigma_i \ell_{[i]}(\boldsymbol{w})$, where $\ell_{[1]}(\cdot) \leq \cdots \leq \ell_{[n]}(\cdot)$. Assume a monotonically increasing probability density function $[0,1] \ni t \to s(t) \in [0,\infty)$. Then $\sigma_i$ can be expressed as $\int_{\frac{i-1}{n}}^{\frac{i}{n}} s(t) dt$. Given a minibatch $i_1, ..., i_m$, SGD computes a stochastic subgradient $g = \sum_{j=1}^{m} \tilde{\sigma}\_{j} \nabla \ell_{i_{[j]}}(\boldsymbol{w})$, where $\tilde{\sigma}\_j = \int_{\frac{j-1}{m}}^{\frac{j}{m}} s(t) dt$ and $\ell_{i_{[1]}}\leq ...\leq\ell_{i_{[m]}}$.  $g$ can potentially be viewed as an unbiased stochastic subgradient of a modified spectral risk $R_{\hat{\sigma}}(\boldsymbol{w})$. The difference between $R_{\sigma}(\boldsymbol{w})$ and $R_{\hat{\sigma}}(\boldsymbol{w})$ is controlled by the minibatch size $m$. Indeed, if $m=n$, then $R_{\hat{\sigma}}(\boldsymbol{w}) = R_{\sigma}(\boldsymbol{w})$. For instance, [1] demonstrates that if $\tilde{\sigma}\_1 = ... = \tilde{\sigma}\_q = 1/q$ and $\tilde{\sigma}\_{q+1} = ... = \tilde{\sigma}\_m = 0$, then $g$ is an unbiased stochastic subgradient of the spectral risk $\frac{1}{q} \sum\_{j=1}^n \gamma\_j \ell_{[j]}(\boldsymbol{w})$, where $\gamma\_j := \frac{\sum\_{l=0}^{q-1} \binom{n-j}{l} \binom{j-1}{m-l-1}}{\binom{n}{m}}$.
> > >
> > > Therefore, even though SGD does not solve the original spectral risk minimization problem, since $g$ is an unbiased stochastic subgradient of a modified spectral risk, the convergence results of vanilla SGD in non-convex settings may be applied. Thus SGD can converge to a stationary point of a modified spectral risk.   For Prospect, [2] claims that the stochastic subgradient estimator it uses is asymptotically unbiased. However, in the non-convex setting, this property may no longer hold. Therefore, the convergence of Prospect can no longer be guaranteed. Even though Prospect may converge to a local minimum point of the original spectral risk, (assume that the coefficient of the strongly concave term is set small enough.) it is challenging to analyze the loss values of the stationary points of Prospect and SGD in non-convex settings, and the analysis is beyond the scope of this paper.
> > >
> > > **References**
> > >
> > > [1] Kawaguchi, K., & Lu, H. (2020). Ordered SGD: A New Stochastic Optimization Framework for Empirical Risk Minimization. International Conference on Artificial Intelligence and Statistics (AISTATS), 669–679.
> > >
> > > [2] Mehta, Ronak, et al. "Distributionally robust optimization with bias and variance reduction." The Twelfth International Conference on Learning Representations, abs/2310.13863, 2024.

---

> > > > ### Comment · Reviewer_fKff · 2024-11-27
> > > >
> > > > Thank you for your answer.  I maintain my current score.

---

### Official Review · Reviewer_CGGB · 2024-11-04

**Soundness:** 3
**Presentation:** 3
**Contribution:** 2
**Rating:** 6
**Confidence:** 3

**Summary:**

This paper proposes a stochastic algorithm for spectral risk minimization with trajectory stabilization for the primal variable. It is claimed that their approach enjoys a near-optimal rate of convergence that matches the known lower bound. They also demonstrate that this method outperforms existing baselines in various experimental settings.

**Strengths:**

This paper is well-written and easy to follow. The claimed contribution is interesting to the community. However, I have a major concern regarding its correctness which I elaborate on in the next section.

**Weaknesses:**

I didn't read the proofs closely but the main theorem seems to be incorrect by a sanity check. To be more specific, $\mu$ is in unit loss/par^2 and $\delta_k$ is in unit loss, but $\delta_k \sim \mu$ in Theorem 1. Moreover, $G$ is in unit loss/par and $\epsilon$ is in unit par^2, so $G/(\mu \sqrt{\epsilon})$ is unitless. However, there is $\log{(G/(\mu^2 \sqrt{\epsilon}))}$ in the sample complexity given in Cor. 1 which makes it not unitless.

Other comments:
1. I do not see error bars (standard errors) reported in all the experiments. Are these results based on a single run or multiple independent runs? Can you report error bars so that we can tell how significant the improvements are?
2. $w^*$ is not defined in Theorem 1.

**Questions:**

See the section above.

---

> ### Author Response · Authors · 2024-11-19
>
> We appreciate the comments provided by reviewer CGGB, especially for pointing out that the complexity in Corollary 1 includes units.
>
> ## 1. Complexity not unitless
>
> We acknowledge that the complexity in Corollary 1 is not unitless.
>
> We point out that the unit in the complexity arises from the use of the inequality $2\left(\mathbb{E}\|\boldsymbol{x}-\overline{\boldsymbol{x}}\|^2\right)^{\frac{1}{2}} \leq \left(1+\mathbb{E}\|\boldsymbol{x}-\overline{\boldsymbol{x}}\|^2\right)$ in the proof of Lemma 2 at line 803. While this inequality is mathematically correct, it results in inconsistent units on both sides of the inequality. To ensure consistent units, we simply need to modify the inequality to $2\left(\mathbb{E}\|\boldsymbol{x}-\overline{\boldsymbol{x}}\|^2\right)^{\frac{1}{2}} \leq \left(D+D^{-1}\mathbb{E}\|\boldsymbol{x}-\overline{\boldsymbol{x}}\|^2\right)$, where $D=G/\mu$. Next, we only need to adjust the coefficients of the affected terms in our analysis and set $\tilde{\delta}_k=D^2\delta_k$ in Theorem 1, where $\delta_k=\min \left(\frac{\mu}{8(k+5)}, \mu(k+1)^{-6}\right)$  is the parameter used in Theorem 1 when we obtained the complexity with units. This allows us to obtain the unitless complexity $O\left(\frac{n^{\frac{3}{2}} G}{\mu \sqrt{\epsilon}} \log \frac{\sqrt{n}}{G \sqrt{\epsilon}}+\frac{L}{\mu} \log \frac{\sqrt{n}}{G \sqrt{\epsilon}} \log \frac{\sqrt{n} G}{\mu \sqrt{\epsilon}}\right)$. Note that the numerator in $\log \frac{\sqrt{n}}{G \sqrt{\epsilon}}$ implicitly includes units of loss. To verify its correctness, as pointed out by reviewer YNq6, $\delta_k$ should be in units of loss (based on Lemma 3). Indeed, $\tilde{\delta}_k$ is in units of loss.
>
> We have uploaded a revised version of the manuscript and corrected our analysis, including Theorem 1 and Corollary 1 in the main text, as well as the proof in Appendix 1. The changes are highlighted in blue. We emphasize that our proof before modification is mathematically correct. To obtain a unitless complexity, we only need to adjust the coefficients of certain terms, and our analysis remains valid.
>
> ## 2. Reporting error bars
>
> In Appendix E of the revised manuscript,  we provide the error bars for the experiments in Section 5.
>
> Specifically, Figure 7 shows the mean training curves and standard deviations of algorithms with minibatching in the linear regression experiments. Since our plots are in log scale, we only keep the upper error bar to make the plots easier to read. Table 4 presents the fairness metrics and their standard deviations from the experiments in Section 5.2. Figure 8 shows the mean training curves, mean worst classification error, and error bars from the experiments in Section 5.3.
>
> We observe that the standard deviations of the fairness metrics in the experiments in Section 5.2 is relatively large. This is because we adopted 5-fold cross-validation, as suggested by [1]. Tables 5-7 and Tables 8-10 present the fairness metrics and suboptimality for each fold of the data. The standard deviations of fairness metrics and suboptimality within each fold is very small, indicating that the standard deviations of results in Section 5.2 mainly stems from differences between folds, rather than the randomness of the algorithms.
>
> We thank the reviewer for highlighting the importance of reporting error bars, and we will integrate these results into the experiments in Section 5.
>
> ## 3. Definition of $\boldsymbol{w}^\star$
>
> We apologize for not formally defining $\boldsymbol{w}^\star$. $\boldsymbol{w}^\star$ should be defined as the optimal solution to Eq. (1). Since the objective function in Eq. (1) is strongly convex under Assumption 1, $\boldsymbol{w}^\star$ is unique.  We have added this definition to Theorem 1 in the revised manuscript.
>
> ## References
>
> [1] Ding, Frances, et al. "Retiring adult: New datasets for fair machine learning." *Advances in neural information processing systems* 34 (2021): 6478-6490.

---

> > ### Comment · Reviewer_CGGB · 2024-11-27
> > **Response to Rebuttal**
> >
> > Thank you for your efforts in putting up the rebuttal. My concerns have been addressed and I have updated my score.

---

### Author Response · Authors · 2024-11-24
**Global Response**

We thank all the reviewers for their constructive and insightful comments and appreciate the time spent on our manuscript. As the discussion stage is nearing its end, we kindly request the reviewers to take some time to review and respond to our rebuttal. We summarize the modifications in our revised manuscript as follows.

- We have corrected Theorem 1 and Corollary 1 to ensure that the complexity in Corollary 1 is unitless. We also revised the corresponding proofs in Appendix A.
- We presented the error bars for the experiments in Section 5 in Appendix E, as suggested by reviewer CGGB.
- In response to reviewers fKff and TvDg's concerns about SOREL's performance in optimizing nonlinear models, we presented the experimental results for training two-layer neural networks in Appendix F.
- We further discussed the dependence of the complexity in Corollary 1 on $n$ in Appendix G, as suggested by reviewers fKff and TvDg.
- We added the definition of $\boldsymbol{w}^\star$ in Theorem 1.
- We revised Appendix B to reduce redundancy with [1], as our experimental setup mainly follows the setup in [1].

A major concern raised by multiple reviewers is that the complexity in Corollary 1 is not unitless. We have corrected this issue in our revised manuscript. We point out that the unit in the complexity arises from the use of the inequality $2\left(\mathbb{E}\|\boldsymbol{x}-\overline{\boldsymbol{x}}\|^2\right)^{\frac{1}{2}} \leq \left(1+\mathbb{E}\|\boldsymbol{x}-\overline{\boldsymbol{x}}\|^2\right)$ in the proof of Lemma 2 at line 803. While this inequality is mathematically correct, it results in inconsistent units on both sides of the inequality. To ensure consistent units, we simply need to modify the inequality to $2\left(\mathbb{E}\|\boldsymbol{x}-\overline{\boldsymbol{x}}\|^2\right)^{\frac{1}{2}} \leq \left(D+D^{-1}\mathbb{E}\|\boldsymbol{x}-\overline{\boldsymbol{x}}\|^2\right)$, where $D=G/\mu$. Next, we only need to adjust the coefficients of the affected terms in our analysis. We emphasize that our proof before modification is mathematically correct. To obtain a unitless complexity, we only need to adjust the coefficients of certain terms, and our analysis remains valid.

We thank all the reviewers again for their insightful feedback. If the reviewers have any further questions or concerns about our work, we are more than happy to provide any clarification.

**Reference**

[1] Mehta, Ronak, et al. "Distributionally robust optimization with bias and variance reduction."  The Twelfth International Conference on Learning Representations, abs/2310.13863, 2024.

---

### Author Response · Authors · 2024-12-01
**Thank you for your feedback**

We would like to express our deepest gratitude to all the reviewers for their acknowledgment of our paper and constructive comments. In particular, we appreciate reviewers CGGB and YNq6 for pointing out that the complexity in Corollary 1 of our initial manuscript was not unitless, and reviewers fKff and TvDg for highlighting the importance of discussing the overall time complexity and the complexity with respect to the sample size $n$.  This has greatly helped us improve our manuscript. Below, we summarize the main concerns raised by the reviewers and how we addressed them.

- Reviewers  CGGB and YNq6 pointed out that the complexity in Corollary 1 of our initial manuscript was not unitless. We clarified that this error was due to an improper use of the inequality at line 803, and that our initial proof was mathematically correct. We have corrected the result and the reviewers have confirmed our result.
- Reviewers fKff and TvDg raised concerns regarding the strongly convex regularizer and whether we are solving the original spectral risk minimization problem. We clarified that the strongly convex regularizer is a regularization term on the model parameters, and that our approach does not involve smoothing the original spectral risk by adding a strongly concave term with respect to $\boldsymbol{\lambda}$, as in some other methods. We have clarified the potential confusion that may arise between the regularizer on the model parameters and the strongly concave term with respect to $\boldsymbol{\lambda}$.
- Reviewers fKff and TvDg raised concerns about the overall time complexity of our method and the complexity with respect to $n$. We further discussed this in Appendix G of our revised manuscript. Our method also has advantages in complexity with respect to the sample size $n$.
- Reviewers fKff and TvDg asked about the performance of our algorithm in the case of nonlinear models or non-convex situations. We presented the experimental results for training two-layer neural networks in Appendix F. Our method achieves significantly lower losses compared to other baselines, even though none of the algorithms have theoretical guarantees.
- Reviewer CGGB pointed out that our experiments in Section 5 lack error bars. We presented the error bars in Appendix E.

We are confident that we have addressed all the reviewers' concerns. The reviewers also acknowledged the presentation of our paper and its contribution to the community. Again, we greatly thank all the reviewers for their constructive comments and the discussions with us.  We hope that the rebuttal phase can be informative for all reviewers.

---

### Meta-Review · Area_Chair_ew7i · 2024-12-21

**Metareview:**

An algorithm called SOREL is proposed for spectral risk minimization, with convergence analysis for convex losses together with a strongly convex regularization. Experiments was limited to linear models in the original submission, but additional ones with nonlinear models are conducted during the discussion. Revision should include the corrected proof and additional experiments.

**Additional Comments On Reviewer Discussion:**

Correctness of the theoretical analysis were discussed between the reviewers and the authors. Reviewers are satisfied with the corrections.

As per request by the reviewers, some additional experiments on nonlinear models are conducted during the discussion.

---

### Decision · Program_Chairs · 2025-01-22

Accept (Poster)